# The unmitigated profile of COVID-19 infectiousness

Ron Sender[1†], Yinon Bar-On[1†], Sang Woo Park[2], Elad Noor[1], Jonathan Dushoff[3,4,5], Ron Milo[1]*

[1]Weizmann Institute of Science, Rehovot, Israel; [2]Department of Ecology and Evolutionary, Princeton University, Princeton, United States; [3]Department of Biology, McMaster University, Hamilton, Canada; [4]Department of Mathematics and Statistics, McMaster University, Hamilton, Canada; [5]M. G. DeGroote Institute for Infectious Disease Research, McMaster University, Hamilton, Canada

**Abstract** Quantifying the temporal dynamics of infectiousness of individuals infected with SARS-CoV-2 is crucial for understanding the spread of COVID-19 and for evaluating the effectiveness of mitigation strategies. Many studies have estimated the infectiousness profile using observed serial intervals. However, statistical and epidemiological biases could lead to underestimation of the duration of infectiousness. We correct for these biases by curating data from the initial outbreak of the pandemic in China (when mitigation was minimal), and find that the infectiousness profile of the original strain is longer than previously thought. Sensitivity analysis shows our results are robust to model structure, assumed growth rate and potential observational biases. Although unmitigated transmission data is lacking for variants of concern (VOCs), previous analyses suggest that the alpha and delta variants have faster within-host kinetics, which we extrapolate to crude estimates of variant-specific unmitigated generation intervals. Knowing the unmitigated infectiousness profile of infected individuals can inform estimates of the effectiveness of isolation and quarantine measures. The framework presented here can help design better quarantine policies in early stages of future epidemics.

*For correspondence: ron.milo@weizmann.ac.il

†These authors contributed equally to this work

Competing interest: The authors declare that no competing interests exist.

## Editor's evaluation

By analyzing a carefully curated dataset of cases observed early, and adjusting for multiple forms of bias, this study provides convincing evidence that in the absence of public health interventions, the duration of infectiousness of COVID-19 (original variant) is longer than previously estimated. These important findings improve our ability to model counterfactual intervention-free scenarios, add to evidence that interventions have reduced the duration of infectiousness, and provide an example of how to navigate the biases and pitfalls inevitably present in outbreak data.

## Introduction

In an emerging epidemic, such as the current COVID-19 pandemic, information about key epidemiological parameters of the causative infectious agent (SARS-CoV-2 in the case of COVID-19) is crucial for monitoring and mitigating the spread of the disease. A central epidemiological parameter, which determines the time scale of transmission, is the generation interval – the time between the infection of the infector (first case) and of the infectee (secondary case). Measuring the generation interval directly is hard in practice, as determining the exact time of infection is challenging. Thus, to infer the generation interval for an emerging infectious disease, researchers usually rely on two widely reported epidemiological parameters: the *incubation period* – the time between infection with the virus and the

**Figure 1.** Definitions of epidemiological time intervals. The incubation period is defined as the time between infection and symptom onset (= $-\alpha_1$ for the infector, $\tau - \alpha_2$ for the infectee). The serial interval (=$\tau$) is defined as the interval between the onset of symptoms of two subsequent transmission events (infector and infectee) and the generation interval is the time lapse between the infections of those individuals (= $\alpha_2 - \alpha_1$). TOST stands for time from onset of symptoms to transmission (*Ferretti et al., 2020b*), and is defined accordingly as the time lapse between symptom onset in the infector and the infection of the infectee (i.e., transmission time). The timeline at the bottom corresponds to the notation used in the Methods section.

onset of symptoms (either for the infector or the infectee) – and the *serial interval* – the time between onset of symptoms of the infector and infectee (*Fine, 2003*; *Svensson, 2007*; *Figure 1*). Key epidemiological delays, such as incubation periods, serial intervals, and generation intervals, vary across hosts and transmission events, and are thus described as distributions rather than fixed values.

The generation-interval distribution plays a key role in determining the spread and control of emerging epidemics such as the ongoing COVID-19 pandemic. At the population level, the generation-interval distribution links incidence of infection, particularly the epidemic growth rate $r$, with the reproduction number $R$ (*Gostic et al., 2020*; *Wallinga and Lipsitch, 2007*). At the individual level, it characterizes the infectiousness profile (i.e., the temporal evolution of infectiousness from the time of infection). In the case of COVID-19, short generation intervals, driven by pre-symptomatic transmission, have limited the effectiveness of different mitigation strategies, including contact tracing (*Ferretti et al., 2020b*), case isolation, quarantine (*Sun et al., 2020*), and testing (*Grassly et al., 2020*; *Johansson et al., 2021*).

The generation- and serial-interval distributions can change over the course of an epidemic. For example, they are affected by the behavior of the population and can be shortened by the introduction of mitigation steps such as social distancing and case isolation, which limit the spread of the disease and reduce the probabilities of transmission after symptom onset (*Ali et al., 2020*). Our study aims to estimate the temporal dynamics of transmissibility of infected cases in the absence of intervention measures, noted hereafter as the 'unmitigated generation interval'. Unbiased estimates of the time profile of transmissibility are important for inferring the effectiveness of self-isolation or quarantine policies in the absence of other interventions.

In practice, estimating the unmitigated infectious profile is expected to be challenging, since even in the absence of any mitigation policies, symptomatic individuals may self-isolate, reducing their own chances of late transmission. To address this issue, we apply a strict data curation procedure to account for which transmission events occurred both before major mitigation steps took place and before awareness of the epidemic became widespread. Most available estimates of the

generation-interval distribution addressed the effects of mitigation only in a limited manner, not fully accounting for steps such as contact tracing and case isolation (*Ferretti et al., 2020a*; *Ganyani et al., 2020*; *He et al., 2020*).

Even after minimizing mitigation and behavioral effects, estimating the generation-interval distribution directly from contact tracing data remains difficult because the timepoint of infection of both the infector and the infectee are usually unknown. Instead, researchers estimate generation-interval and incubation period distributions by calculating the likelihood of observing all serial intervals in the transmission pair dataset (*Ferretti et al., 2020a*; *Ferretti et al., 2020b*; *He et al., 2020*) or else, they simply use the serial-interval distribution as a proxy for the generation-interval distribution (*Flaxman et al., 2020*).

While the serial-interval-based framework has been widely applied to infer the generation-interval distribution of COVID-19 (*Ferretti et al., 2020a*; *Ferretti et al., 2020b*; *Ganyani et al., 2020*; *He et al., 2020*; *Sun et al., 2020*), there are several key methodological issues that could lead to considerable biases. First, the distribution of realized serial intervals depends on the rate of the spread of the disease as well as the direction from which they are measured: either forward from a cohort of infectors who developed symptoms at the same time or backward from a cohort of infectees (*Park et al., 2021*). A cohort of individuals that develop symptoms on a given day is a sample of all individuals who have been previously infected. When the incidence of infection is increasing, recently infected individuals represent a bigger fraction of this population and thus are over-represented in this cohort. Therefore, we are more likely to encounter infected individuals with a short incubation period in this cohort compared to an unbiased sample. The forward serial interval is calculated for a cohort of infectors who developed symptoms at the same time and therefore is sensitive to this bias. These dynamical biases are demonstrated using epidemic simulations by *Park et al., 2020*. However, most analyses of serial-interval distributions assume that the incubation periods of the infector and infectee follow the same distribution (*Ganyani et al., 2020*; *He et al., 2020*; *Sun et al., 2020*), and only a few studies partially account for this dynamical bias (*Ferretti et al., 2020a*; *Ferretti et al., 2020b*). Second, incubation periods and temporal profile of infectiousness are likely to be correlated across infectors – that is, individuals that show symptoms later or earlier are also more likely to infect others later or earlier, respectively. Most available studies make strict assumptions on the relationship between the incubation period and the generation interval – either assuming that they are independent (*Ferretti et al., 2020b*; *Ganyani et al., 2020*; *Sun et al., 2020*) or that the time from onset of symptoms to transmission (TOST) is independent of the incubation period (*He et al., 2020*). Only a few studies have compared various correlation models (*Ferretti et al., 2020a*) or explicitly modeled the infectiousness profile relative to the incubation period (*Hart et al., 2021*). Finally, biases can arise from the data collection process. For example, determining who infected whom based on their symptom-onset dates can miss pre-symptomatic transmission. Likewise, long serial intervals may represent multiple chains of transmissions where intermediate hosts were not correctly identified. These biases can cause overestimation of the mean serial interval as well as the mean generation interval.

Currently, no available estimate for the generation interval deals with all the biases described above, impairing our ability to accurately describe the infectiousness of SARS-CoV-2-infected individuals in the absence of interventions. Here, we aggregate all available transmission data for Wuhan, China, in the initial stages of the pandemic, when the effects of mitigation steps were minimal, and employ a statistical framework that addresses the major sources of bias in estimating the generation-interval distribution. We estimate a median generation interval of 7.9 days (95% confidence interval [CI] 6.8–9) and an average of 9.7 days (95% CI 8.3–11.2), suggesting that the infectious period is much longer than previously thought. We further combine our generation-interval estimates with previously inferred viral load trajectories (*Kissler et al., 2021*; *Hay et al., 2022*) to extrapolate unmitigated generation-interval distributions of alpha, delta, and omicron variants. The estimated unmitigated generation-interval distribution could be adopted for answering questions about quarantine and isolation policy, as well as for estimating the original $R_0$ at the initial spread in China. However, estimation of instantaneous $R(t)$ should account for changes in generation-interval distributions, reflecting mitigation effects and the current variant.

## Results

We estimated the unmitigated generation interval by focusing on the first period of transmission in China, thus minimizing the potential impacts of early interventions. To choose our analysis period, we relied on previous analyses of the early outbreak and the timeline of interventions in Wuhan and mainland China. We quantified the forward serial-interval distributions based on the symptom-onset dates of the infector. We found that the mean forward serial interval stayed constant until around the January 17, 2020, and then decreased gradually, indicating changes in transmission dynamics (*Figure 2*). In particular, a strict restrictions on mobility (lockdown) imposed in Wuhan city on January 23 (*Kraemer et al., 2020*) likely impacted generation (and therefore serial) intervals of infectors who were infected a few days prior to this date – for example, an individual who developed symptoms on January 23 would have had reduced transmission after January 23, thereby shortening their generation and serial intervals. The clear negative trend in the mean serial interval from January 17–18 onward also matches the timing of the decrease in the effective reproduction number *R(t)* for domestic cases in Wuhan, China, estimated by *Lipsitch et al., 2020*. Therefore, we chose January 17 as our cutoff date. Large uncertainties in early serial-interval data limited our ability to detect changes in the mean forward serial interval before January 17. Previous studies *Kraemer et al., 2020*; *Lipsitch et al., 2020*; *Park et al., 2021* found no clear signs of change in the growth of the epidemic prior to the period between the 16th and the 19th of January.

We used the transmission pairs for which the infector developed symptoms between December 12, 2019, and January 17, 2020, as our main dataset for estimating the unmitigated generation-interval distribution. This dataset includes a total of 77 transmission pairs with a mean serial interval of 9.1 days (95% CI: 7.9–10.2), and a standard deviation of 5.2 days. Although this is substantially longer than the mean of 7.8 days (95% CI: 7–8.6 days) suggested by *Ali et al., 2020*, for the early period of the epidemic, there is considerable uncertainty in both estimates with overlapping CI. Nonetheless, a lower mean serial interval estimated by Ali et al. likely reflects their decision to include infectors who developed symptoms up to January 22, who were already subject to effects of mitigation strategies. Other studies that did not differentiate different stages of the epidemic estimated a much lower mean serial interval (4–6 days) (*He et al., 2020*; *Sun et al., 2020*; *Zhang et al., 2020*; *Zhanwei et al., 2020*).

We inferred the unmitigated generation-interval distribution of SARS-CoV-2 transmission based on an integrative curated dataset, which focuses on the early-outbreak period in China.

We used the maximum likelihood framework to estimate the parameters of the joint bivariate distribution of the generation interval and the incubation period, assuming a known incubation period distribution (*Xin et al., 2021*) with a mean of 6.3 days and a standard deviation of 3.6 days. We estimate that the unmitigated generation-interval distribution has a median of 7.9 days (95% CI: 6.8–9), a mean of 9.7 (95% CI: 8.3–11.2) days and standard deviation of 6.9 (95% CI: 4.3–10.1) days. Furthermore, we estimate a correlation parameter (see Methods) of 0.75 (95% CI: 0.5–0.9). Our estimates are robust to the choice of data sources used in the analysis included (*Appendix 1—figure 4*).

We note that our estimated mean generation interval is longer than the observed mean serial interval (9.1 days) of the period in question. This is supported by the theory (*Park et al., 2021*) of the dynamical effects of the epidemic – in contrast to the common assumption that the mean generation and serial intervals are identical. During the exponential growth phase, the mean incubation period of the infectors is expected to be shorter than the mean incubation period of the infectee – this effect causes the mean forward serial interval to become longer than the mean forward generation interval of the cohorts that developed symptoms during the study period. However, these cohorts of infectors with short incubation periods will also have short forward generation (and therefore serial) intervals due to their correlations. When the latter effect is stronger, the mean forward serial interval becomes shorter than the mean intrinsic generation interval, as these findings suggest.

The joint bivariate distribution and its marginal distributions are shown in *Figure 3A*. With or without the growth rate adjustment, the model was able to fit the observed serial-interval data well (*Figure 3B*). Using the inferred bivariate distribution, we derived the distribution of TOST, as shown in *Figure 3—figure supplement 1*. The negative side of this distribution gives the pre-symptomatic transmission, which constitutes ≈20% (95% CI: 6–32%) of total transmission.

A comparison with the current available estimates of the generation-interval distribution (*Ferretti et al., 2020a*; *He et al., 2020*; *Sun et al., 2020*) reveals that the inferred distribution has a heavier (right) tail (*Figure 4*) and a higher median (7.9 days compared to 5.4–5.8 days) and standard deviation

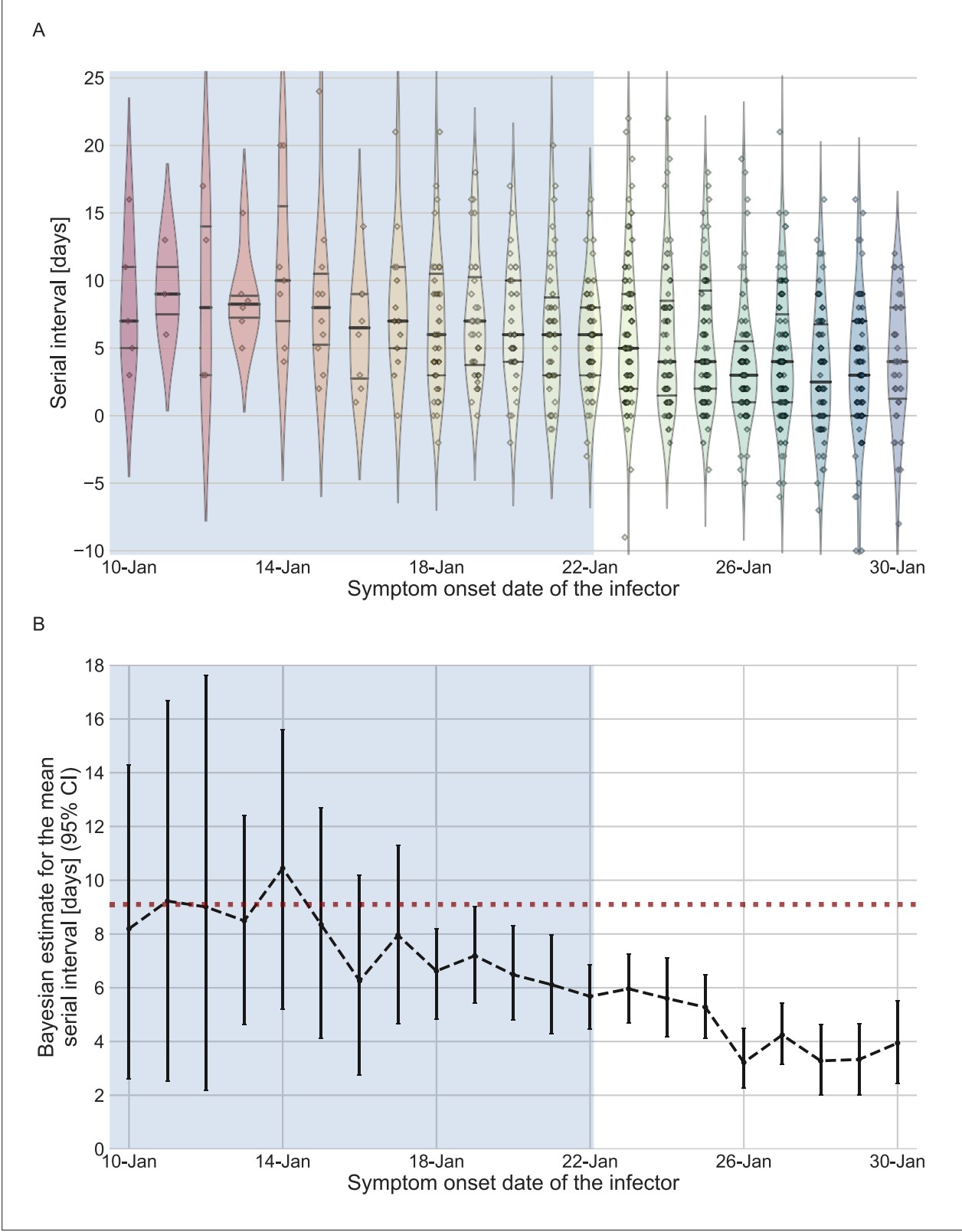

**Figure 2.** The serial-interval dataset and the estimates of its mean during the early-outbreak period. (A) The empirical distributions of forward serial intervals in the combined dataset, grouped based on the symptom-onset dates of the infectors and visualized using a violinplot. For pairs with uncertainty regarding the exact dates of symptom onset, we used a date in the middle of the uncertainty range. The violin shapes represent a kernel density estimation of the underlying distribution. The median and interquartile range (percentiles 25–75) are presented using dotted horizontal lines

*Figure 2 continued on next page*

*Figure 2 continued*

within the shape. The diamonds represent the data points for each of the dates of infector symptom onset. The dataset contains transmission pairs with infectors who developed symptoms from December 12 onward. Dates prior to January 10 are not shown as the data are too sparse. (B) The estimates of the mean serial interval, based on a parametric Bayesian inference (see Supplementary Information for details). The error bars represent the 95% confidence interval (CI) of the estimates. The dashed horizontal line represents the observed mean serial interval for the period up to January 17. Dates up to January 17, 2020, are highlighted in both panels as they represent the period of unmitigated transmission.

(6.9 days compared to 3.3–3.9 days). For example, the gamma distribution assumed by *Johansson et al., 2021*, to give an infectious period of about 10 days (and a peak at 5 days) for the analysis of quarantine and isolation policies has a far smaller tail. One way to quantify the difference in the tails of the different estimates is by comparing the proportion of transmission after a certain timepoint. When comparing the proportion of transmission after day 14, there are clear differences from previously reported distributions. The distributions of Ferretti et al., He et al., and Sun et al. indicate a residual fraction of transmission after 14 days of 2–4%, while the distribution assumed by Johansson et al. indicates only 0.2%. In contrast, our inferred generation-interval distribution predicts that about 18.5% (95% CI of 10–25%) of the transmission occurs after 14 days in the unmitigated scenario (assuming the behavior doesn't change due to quarantine, isolation, testing, etc.).

In addition to the possible dynamical and statistical biases considered in our analysis, the resulting wide generation-interval distribution might be affected by biases in the data collection process as detailed in the Introduction and Methods sections. The estimated generation-interval distributions were sensitive to the cutoff date with an estimated median of 6.5–8 days and estimated means of 7–10 days for periods ending on January 16 to January 19, 2020 (*Figure 5A–C* and *Figure 5—figure supplement 1*). Using cutoff dates of January 21 or later gives generation-interval distribution with median of less than 6 days, and residual transmission of 5% at 14 days after infection, similar to the values found in previous sources (*Ferretti et al., 2020a*; *He et al., 2020*; *Sun et al., 2020*). This demonstrates the impact of mitigations in biasing the inference of generation-interval distributions.

Switching the order of some of the transmission pairs caused a decrease in both the median and mean of the generation interval, as well as a decrease in the correlation parameter (*Figure 5G–I*, *Figure 5—figure supplement 2*).

The sensitivity analysis to high serial-interval values caused a slight decrease in the mean generation interval, but still resulted in a wide distribution. Removing the transmission pairs with the highest serial intervals from the dataset caused a small decrease in the generation-interval distribution. For a removal of the top 10% values, the inferred distribution has a median of 7.2 days and a mean of 8.3 days (*Figure 5D–F*, *Figure 5—figure supplement 3*). As switching the direction of transmission among randomly selected infector-infectee pairs gives negative serial intervals (and thus lower mean serial interval), a decrease in the mean generation-interval distribution was expected. However, even when reordering 10% of the pairs the distribution is wide: for example, the median of bootstrap estimates for the median generation interval is 7.2 days (*Figure 5H*). These bootstrap estimates also yield substantial residual transmission at 14 days (*Figure 5I*).

Other factors of uncertainty in the estimate are the growth rate and incubation period distribution we assume for the inference of the distribution. Changing the assumed growth rate during this period had very little effect on the results, with estimated mean increasing from 9.5 to 9.7 days, as assumed growth rates decreased from 0.16 to 0.04 day$^{-1}$ (*Appendix 1—figure 7*). Changing the incubation period to one with a median in the range of 4–5.5 days (*Appendix 1—figure 10*), as well as inclusion of severe cases in the dataset (*Appendix 1—figure 11*) had very little effect. These sensitivity analyses demonstrate the robustness of our conclusion: the unmitigated generation-interval distribution is likely wider than previously thought.

To quantify the effect of our estimated generation-interval distribution on the estimates of the basic reproduction number $R_0$ of SARS-CoV-2 wild type, we use the growth rate estimated in a recent study of the early outbreak dynamics in China (*Tsang et al., 2020*). Combining our estimated generation-interval distribution with the early growth rate (*Wallinga and Lipsitch, 2007*), we find $R_0$ to be 2.2 with a CI of 1.9–2.7 (*Appendix 1—figure 5*).

Finally, we estimated the unmitigated generation-interval distributions for new SARS-CoV-2 variants by incorporating viral kinetic information (*Figure 6*). The median generation interval of both the alpha and delta variants was estimated to be 6.7 days, 15% shorter than the original variant. The

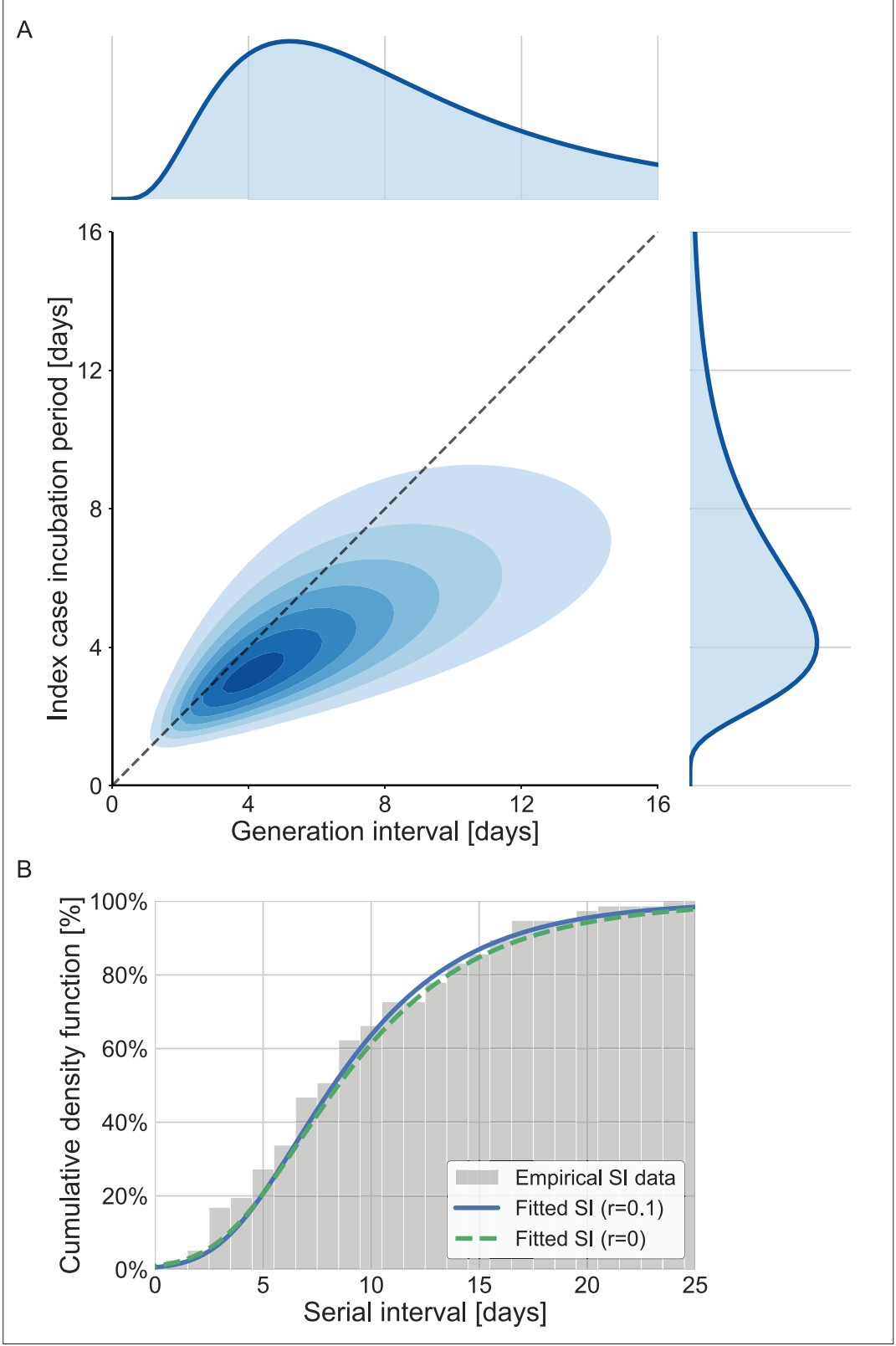

**Figure 3.** The joint distribution of generation interval and incubation period. Representations of the inferred joint distribution results are based on maximum likelihood analysis. (**A**) The joint bivariate distribution (bottom left graph), shown as contours over the plane of generation intervals (x-axis) and incubation period distribution (y-axis). The correlation parameter (in log space, see Methods) was found to be 0.75 (0.5–0.9 95% confidence interval [CI]).

*Figure 3 continued on next page*

*Figure 3 continued*

The panel also shows the univariate components of the joint distribution: the generation-interval distribution (top graph, sharing the same x-axis) and the incubation period distribution (bottom right graph, sharing the same y-axis). The incubation period distribution was assumed to follow a lognormal distribution with a shape parameter of 0.53 and a scale parameter of 5.5 days, following *Xin et al., 2021*. The dashed diagonal line describes equal incubation period and generation interval (time from onset of symptoms to transmission [TOST] equal to zero). Left of this line could be found the pre-symptomatic fraction of transmission. (**B**) Cumulative histogram of the empirical serial intervals and the parametric distribution derived from the maximum likelihood joint distribution. The estimated serial-interval distribution was derived using the likelihood calculation given the reported growth rate of $r$=0.1 day$^{-1}$ (*Tsang et al., 2020*). For comparison the dashed line represents the intrinsic serial interval distribution, estimated by *Equation (2)* with the parameters derived from the maximum likelihood analysis (corresponding to the case of $r$=0 day$^{-1}$).

The online version of this article includes the following figure supplement(s) for figure 3:

**Figure supplement 1.** The distribution of time from onset of symptoms to transmission (TOST).

median generation interval of the omicron variant was estimated to be 5.8 days, ≈30% shorter than the original variant. Even though these generation intervals are considerably shorter than the 7.9 days median generation intervals of the wild type, there might be a considerable amount of late transmission – for example, we estimate that more than 15–19% of the transmission potential occurs 5 days after symptoms for the three variants of concern (VOCs), in the absence of mitigation.

## Discussion

In this work, we assembled transmission pair data from 12 datasets representing the early-outbreak period in China, and modeled the relationship between disease transmission and symptom onset using a bivariate lognormal distribution. By applying a maximum likelihood framework, we found that the unmitigated generation-interval distribution has a heavier right tail than previously estimated (*Ferretti et al., 2020a*; *He et al., 2020*; *Sun et al., 2020*), corresponding to a larger mean and standard deviation. The bias in the previous estimates likely reflects the effects of mitigation steps, such as quarantine of exposed individuals, as well as changes in awareness-driven behavior, such as faster self-isolation after symptom onset, that prevent transmission during late stages of infection. These sources of bias were not fully accounted for in previous estimates, leading to substantial underestimation of the generation-interval distribution.

Our sensitivity analysis of the cutoff date for the period of unmitigated transmission indicates that using late cutoff dates, such as the January 21, leads to similar underestimation as seen in the previous sources. However, these dates correspond to periods when transmission dynamics were affected heavily by mitigations, such as the Wuhan lockdown that started on January 23. Therefore, our results, based on an earlier cutoff date, are more representative of the unmitigated scheme.

Superspreading events are considered an important feature of the spread of COVID-19. Indeed, if the transmission pair dataset comprised a large number of cases from a single event, the inferred infectiousness profile would be biased due to strong statistical dependencies. However, the data we chose to include in the analysis consisted of at most two events with more than two infectees (4 and 7), and therefore superspreading likely had negligible effects on our analysis.

Accounting for potential correlations between the incubation period and the generation interval provided a better estimate of the proportion of pre-symptomatic transmission. Our results suggest that, on average, only ≈20% (6–32%) of the unmitigated transmission happens before symptoms appear, lower than commonly stated values that already include mitigation effects (40–60% *Ferretti et al., 2020a*; *Sun et al., 2020*). When mitigation strategies are introduced, we would expect the amount of post-symptomatic transmission to decrease, leading to an increase in the fraction of pre-symptomatic transmission. Thus, it is not surprising that our estimate of the proportion of pre-symptomatic transmission is lower than previous estimates that looked at a later period (*Ferretti et al., 2020a*; *Sun et al., 2020*). Furthermore, our results match the trend shown by *Sun et al., 2020*, in which the faster isolation of cases increases the pre-symptomatic fraction of transmission and shortens the mean generation interval.

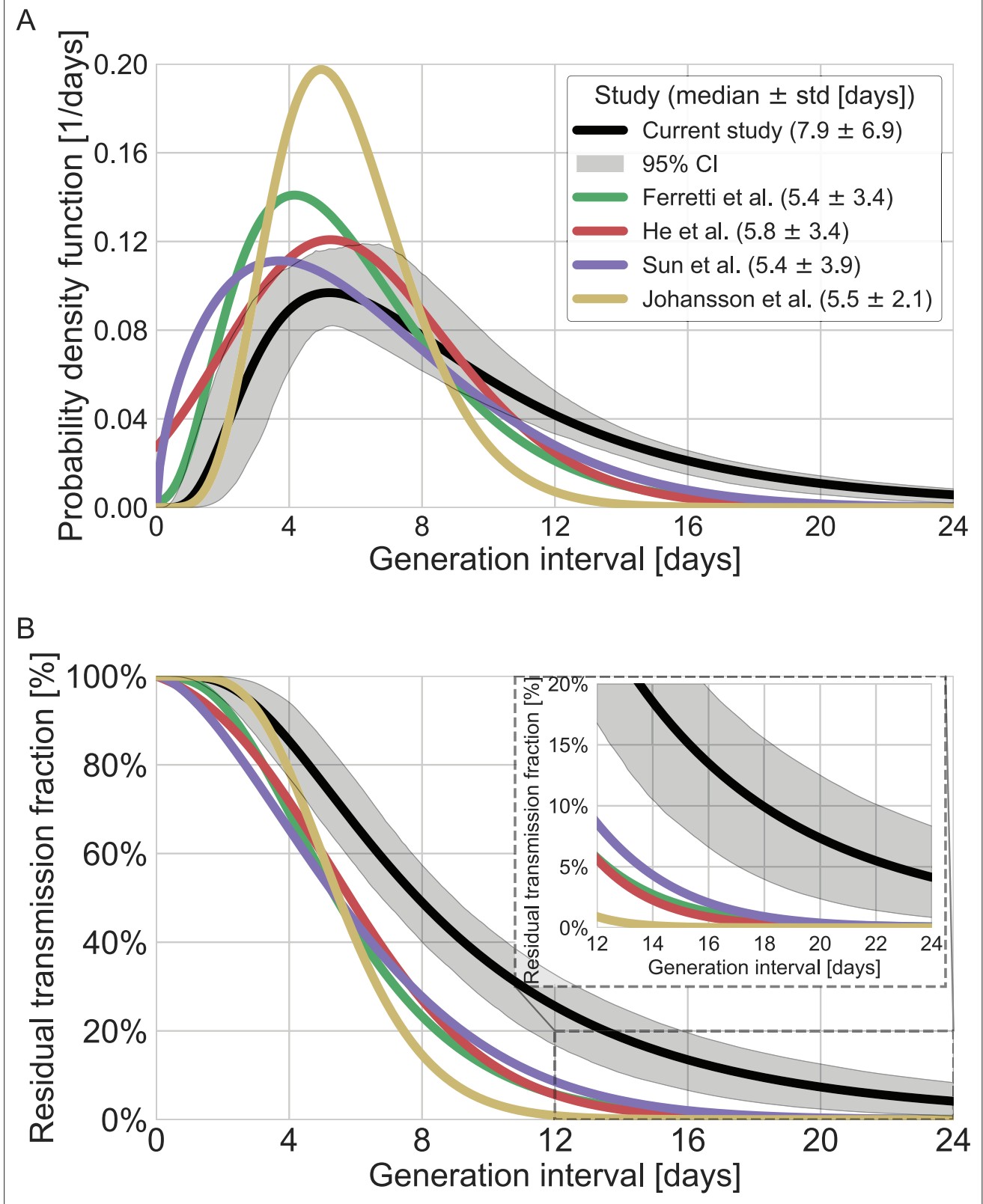

**Figure 4.** Comparison of the mean generation-interval distribution with those of previous studies. The generation-interval distribution inferred by maximum likelihood presented alongside available estimates from the literature (*Ferretti et al., 2020a*; *He et al., 2020*; *Johansson et al., 2021*; *Lauer et al., 2020*; *Sun et al., 2020*). (**A**) The probability density functions of the distributions. The legend reports the median and standard deviation of each of the distributions. (**B**) The survival function of the generation-interval distribution, defined as the complement of the cumulative distribution,

*Figure 4 continued on next page*

*Figure 4 continued*

representing the residual fraction of transmission after a designated time since infection. The inset shows a zoom-in on the period of 12–24 days after exposure, a period in which there is a substantial difference between the current estimate and those from previous studies. The highlighted area represents the 95% confidence interval of the maximum likelihood estimate.

The online version of this article includes the following figure supplement(s) for figure 4:

**Figure supplement 1.** The residual transmission accounting for self-isolation.

To check whether these results are sensitive to our choice of using a bivariate lognormal distribution to characterize the joint distribution of the generation interval and the incubation period, we repeated our analysis using a different functional form using an adjusted logistic TOST model following *Ferretti et al., 2020a* (see supplementary for details). Both models estimate large means and standard deviations of the generation intervals, and a low proportion of pre-symptomatic transmission for the current dataset. Applying both models to the data from *Ferretti et al., 2020a*, produced similar distributions with lower estimates for the mean generation interval and higher per-symptomatic proportion (*Appendix 1—figure 6*). This indicates that the results presented in this study are a product of the focus on the data prior to mitigation steps, in combination with the correction for the growth of the epidemic.

Following the sensitivity analyses to the cutoff date, the growth rate, and the model of infectiousness, we can see which of the three biases described in *Table 1* has the greatest effect. We conclude that the cutoff date seems to be the dominant factor in our analysis, presumably meaning that taking the effects of interventions into account is the most important for an accurate estimate of the generation-interval distribution. Additional sensitivity analyses, such as to the assumed incubation period, also support this conclusion, as they show only a minor effect.

Our analysis relies on datasets of transmission pairs gathered from previously published studies and thus has several limitations that are difficult to correct for. Transmission pairs data can be prone to incorrect identification of transmission pairs, including the direction of transmission. In particular, pre-symptomatic transmission can cause infectors to develop symptoms after their infectees, making it difficult to identify who infected whom. Data from the early outbreak might also be sensitive to ascertainment and reporting biases which could lead to missing links in transmission pairs, causing serial intervals to appear longer (e.g., people who transmit asymptomatically might not be identified). Moreover, when multiple potential infectors are present, an individual who developed symptoms close to when the infectee became infected is more likely to be identified as the infector. These biases might increase the estimated correlation of the incubation period and the period of infectiousness. We have tried to deal with these biases by using a bootstrapping approach, in which some data points are omitted in each bootstrap sample. The relatively narrow ranges of uncertainty suggest that the results are not very sensitive to specific transmission pairs data points being included in the analysis. We also performed a sensitivity analysis to address several of the potential biases such as the duration of the unmitigated transmission period, the inclusion of long serial intervals in the dataset, and the incorrect orderings of transmission pairs (see Methods). The sensitivity analysis shows that although these potential biases can decrease the inferred mean generation interval, our main conclusions about the long unmitigated generation intervals (high median length and substantial residual transmission after 14 days) remained robust (*Figure 5*). Due to the nature of early spread of a new unknown disease, it is nearly impossible to find two completely unrelated datasets from the period prior to mitigation, limiting the ability of further validation of the current results.

Our estimates of the unmitigated generation-interval distribution can inform quarantine policy. The tail of the survival function (*Figure 4B*) indicates that individuals infected with the wild type still have, on average, ≈18% of their transmission potential 14 days after infection. We also found a strong correlation of the incubation period with the generation interval, accentuating the importance of quickly isolating individuals as soon as they show symptoms.

Determining the optimal period of quarantine for individuals exposed to COVID-19 is hard, as it needs to balance the prevention of further transmission with personal and economic costs of longer quarantine. It is important to consider the basic risk of transmission underlying those considerations, by looking at the distribution of infectiousness in the absence of mitigation measures. *Johansson et al., 2021*, estimates for the residual transmission across different quarantine policies (e.g., with and without testing before release) have served as the basis for recent recommendations by the U.S.

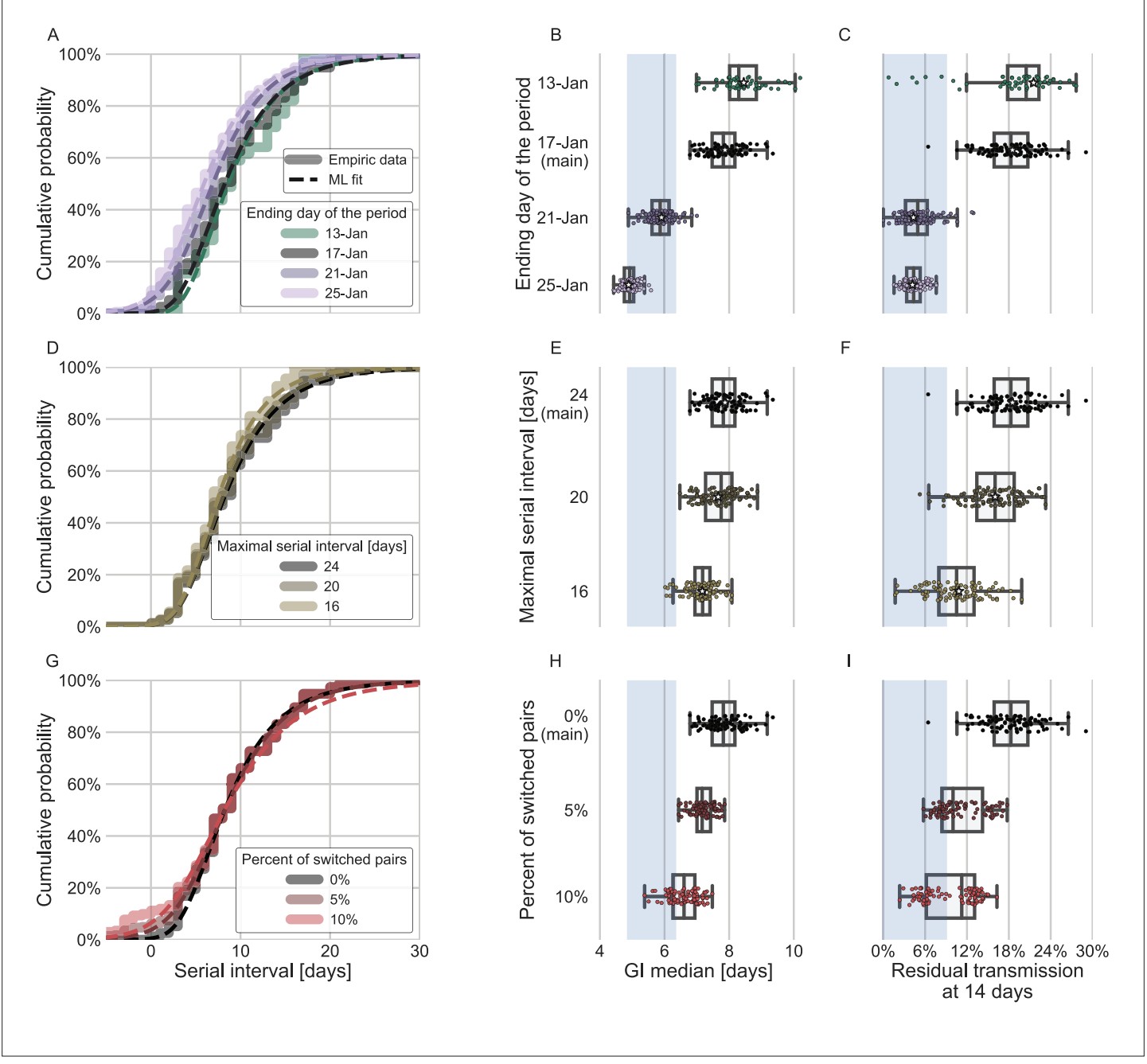

**Figure 5.** Sensitivity analyses of the inferred generation interval. A comparison of the results of sensitivity analysis to three factors: the period chosen to represent the unmitigated transmission (**A–C**), the inclusion of the longest serial intervals in the dataset (**D–F**), and the ordering of the transmission pairs (**G–I**). (**A, D, G**) Cumulative histogram of the empirical serial intervals and the parametric distribution derived from the maximum likelihood joint distribution. The estimated serial interval distribution was derived using the likelihood calculation given the reported growth rate of $r=0.1$ day$^{-1}$ (**Tsang et al., 2020**). (**E, H**) Best estimates and distributions of the resulting median of the inferred generation-interval distribution. A black star marks best estimates. Ranges are given as boxplots. The box represents the interquartile range (percentiles 25–75) and the whiskers represent the maximal range of the distribution apart from outliers (defined as data points exceeding the interquartile range by a factor of 1.5). Each dot represents a single bootstrapping iteration. The blue shaded region represents the values from previous studies (**Ferretti et al., 2020a**; **He et al., 2020**; **Sun et al., 2020**; **Tsang et al., 2020**). (**F, I**) Best estimates and distributions of the resulting residual transmission at 14 days since infection derived from the inferred generation-interval distribution. The best estimates and ranges are shown in the same manner as the distribution parameters in panels **E, H**.

The online version of this article includes the following figure supplement(s) for figure 5:

**Figure supplement 1.** Sensitivity analysis regarding the choice of period for analysis.

*Figure 5 continued on next page*

*Figure 5 continued*

**Figure supplement 2.** Estimates for the bivariate incubation period and generation-interval distribution were obtained for adjusted datasets in which the order of transmission was switched between the infector and infectee (giving a negative serial interval).

**Figure supplement 3.** Sensitivity analysis to the top values of serial intervals.

Centers for Disease Control and Prevention (**CDC, 2020**) for a 10-day quarantine period (without PCR testing) for exposed individuals. As can be seen in **Figure 4B**, our results suggest that this analysis underestimates the residual transmission after 10 days by an order of magnitude for the average individual (35% of the transmission vs. 4%). One of the first and ongoing policies from mitigating transmission is mandatory self-isolation for individuals developing COVID-19-related symptoms (**Johansson et al., 2021**; **Quilty et al., 2021**). We estimate a strong correlation of incubation period and infectiousness, enhancing the contribution of self-isolation to transmission prevention. However, even when considering self-isolation of 70% of individuals immediately upon symptoms, as **Johansson et al., 2021** assumed in their analysis, we still find a residual transmission of 11.8% compared to 1.3% in Johansson et al.'s estimates (**Figure 4—figure supplement 1**). Indeed, the unmitigated infectiousness profile suggests that without testing, the residual transmission after quarantine would be substantially higher – thus supporting the policy of requiring PCR or rapid tests for ending quarantines, as required in many countries. The current study does not analyze the possible benefits of such testing policies directly, but only of self-isolation by individuals who developed COVID-19 symptoms. In addition, quarantine and isolation measures typically begin several days after the infection event, suggesting that the actual amount of post-quarantine and isolation transmission would be lower than what we estimate.

The basic reproduction number $R_0$ estimates derived here are close to reported values from early in the epidemic value (**Chinazzi et al., 2020**; **Imai et al., 2020**; **Li et al., 2020**; **Wu et al., 2020b**), despite the longer estimate for the generation-interval distribution. This is mainly due to using the corrected growth rate, which is considerably lower than previously assumed values (**Tsang et al., 2020**).

SARS-CoV-2 viral load trajectories serve an important role in understanding the dynamics of the disease and modeling its infectiousness (**Quilty et al., 2021**; **Cleary et al., 2021**). Indeed, the general shapes of the mean viral load trajectories and culture positivity, based on longitudinal studies, are comparable with our estimated unmitigated infectiousness profile (**Figure 6—figure supplements 1 and 2**, comparison with **Chu et al., 2022** ; **Killingley et al., 2022**; **Kissler et al., 2021**). However, the nature of the relationship between viral load, culture positivity, symptom onset, and real-world infectivity is complex and not well characterized. Therefore, the ability to infer infectiousness from viral load data is very limited, especially near the tail of infectiousness, several days following symptom onset and peak viral loads. Viral load models are usually made to fit the measurements during an initial exponential clearance phase and in many cases miss a later slow decay (**Kissler et al., 2021**). Furthermore, there is considerable individual-level variation in viral trajectories that isn't accounted for in population-mean models (**Kissler et al., 2021**; **Singanayagam et al., 2021**). Other factors limiting the ability to compare generation-interval estimates with viral loads models are the variability of the incubation periods and its relation to the timing of the peak of the viral loads, and the great uncertainty and apparent non-linearity of the relation between viral loads and culture positivity (**Jaafar et al., 2021**; **Jones et al., 2021**). Due to these caveats and in order to avoid over-interpretation of viral load data, we restrict our extrapolation of new VOCs' infectiousness to a single parameter characterizing the viral duration of clearance.

New SARS-CoV-2 VOCs continue to emerge and replace previous lineages, adding uncertainty to the pandemic transmission dynamics, including the shape of infectiousness profiles. Although a few studies have tried to characterize generation- and serial-interval distributions of new variants (**Hart et al., 2021**; **Pung et al., 2021**; **Ryu et al., 2021**; **Hart et al., 2022**), these analyses are necessarily subject to behavioral and intervention effects and are likely to underestimate the true duration of SARS-CoV-2 infectiousness. Instead, we estimated the unmitigated infectiousness profiles of new variants by comparing the differences in decay rates of viral load trajectories (**Hay et al., 2022**; **Kissler et al., 2021**). Our analyses suggested that the unmitigated generation intervals of the alpha and delta variants are shorter by 15% than those of the original strain and by 30% for the omicron variant (**Figure 6**). Our estimates of residual transmission (more than 15% transmission occurring at least 5 days after symptom onset for the alpha, delta, and omicron variants) suggest that caution is needed

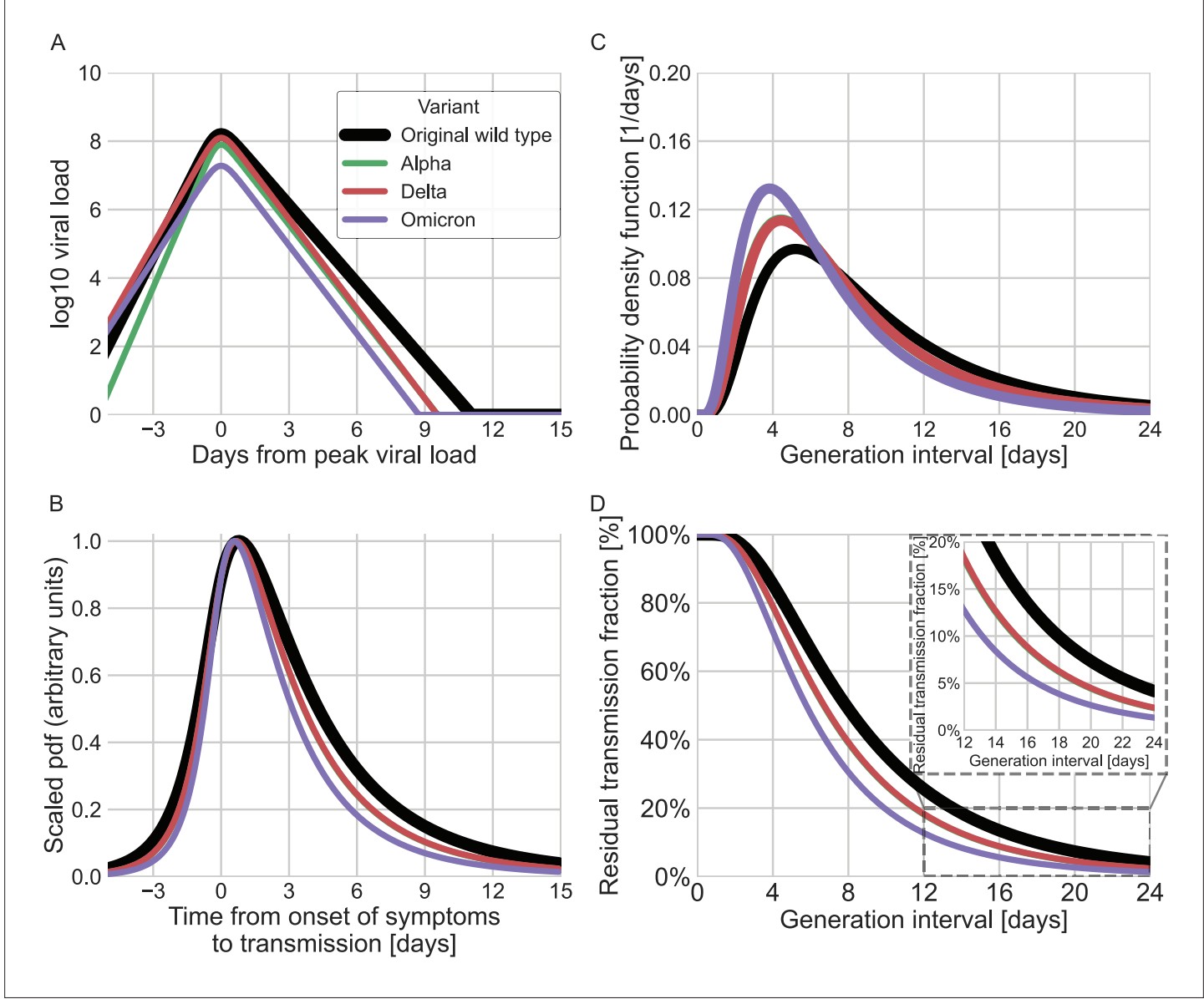

**Figure 6.** Using viral load trajectory for modeling other variants of concern (VOCs). (A) The mean viral load trajectory for the main VOCs according to Kissler et al. (alpha and delta) and Hay et al. (delta and omicron). (B) Infectiousness profiles as derived from the probability density functions of the time from onset of symptoms to transmission (TOST) distribution (scaled to their maximal values). The black curve was derived for the original Wuhan variant (assumed to be close to non-VOCs in Kissler et al. study) using a maximum likelihood inference. The profiles for the alpha, delta, and omicron variants were estimated by scaling the time of the distribution by the ratio of the clearance's durations. (C) The probability density function of the generation-interval distribution extrapolated for the various variants. (D) The survival function of the generation-interval distribution extrapolated for the various variants. The inset shows a zoom-in on the period of 12–24 days after exposure. The extrapolated distributions for the alpha and delta variants are extremely close, hence the green line is hidden by the red line in panels B–D.

The online version of this article includes the following figure supplement(s) for figure 6:

**Figure supplement 1.** Variability of the viral loads and the potential connection with inferred *infectiousness*.

**Figure supplement 2.** Comparison of estimated infectiousness profile with viral loads and positive culture data.

with the CDC's newly updated (**CDC, 2021**) isolation guideline in the absence of additional measures, such as testing, before releasing isolated individuals. Our extrapolations are necessarily crude given the complex relationship between viral load, symptomaticity, and infectiousness discussed above. Moreover, compartmentalization in the respiratory tract, aerosolization, receptor binding affinity, and immune history can also play important roles in determining the infectiousness profiles of SARS-CoV-2

**Table 1.** The main biases of infectiousness profile inference from serial-interval data discussed in the Introduction section.

| Source of bias | Expected net effect on the inferred generation-interval distribution | Current study's approach for correcting the bias | Studies who considered this bias |
|---|---|---|---|
| Mitigation steps and awareness limit the spread of the disease | Underestimation of the mean generation interval | Curation of cases focus only on early spread | – |
| Realized serial intervals depends on the rate of the spread of the disease | Systematic difference between serial- and generation-interval distributions | Correction for backward incubation period distribution | *Park et al., 2021*; *Ferretti et al., 2020b* |
| Possible correlation of incubation periods and temporal profile of infectiousness | Underestimation of the mean generation interval | Modeling infectiousness using incubation period and generation-interval bivariate distribution | *Hart et al., 2021*; *Ferretti et al., 2020a* |

variants (*Puhach et al., 2022*). Furthermore, changes in population-level susceptibility can also cause the generation-interval distribution to change over time (*Kenah et al., 2008*; *Champredon and Dushoff, 2015*).

Our analysis focuses on estimating unmitigated intervals. Therefore, our estimates don't take into account the effect of current interventions and behavioral changes. Nonetheless, these estimates can be useful for assessing isolation and contact tracing measures. One prediction of our extrapolation procedure is that the durations of infectiousness and incubation periods for the alpha and delta variants would be shorter by 15% relative to the original strain, which is supported by independent studies (*Brandal et al., 2021Grant et al., 2021*; *Hwang et al., 2021*; *Singanayagam et al., 2021*; *Hart et al., 2022*). Further transmission data as well as rich viral load trajectory data could assist in better inferences of the infectiousness profiles of new variants.

The current analysis provides an updated benchmark for the unmitigated profile of SARS-CoV-2 infectiousness. Furthermore, with the emergence of new VOCs, which may exhibit altered transmission dynamics than previously dominant wild type (*Kissler et al., 2021*), future studies could use our framework to update estimates of the generation-interval distributions for these emerging strains even under mitigation conditions and with inference of the correlation to the incubation period.

Taken together, our results demonstrate the importance of considering possible biases in the serial-interval data used for estimating the generation-interval distribution, as well as the underlying assumptions made when estimating the distribution from the source data. Our analysis provides a view of the infectiousness profile of an infected individual in absence of mitigation steps, which is a key ingredient of many models used for guiding policy.

## Methods
### Data collection

Data on serial intervals of transmission events were gathered from published and preprint literature, using a literature survey as described in the supplementary information. In order to control for biases introduced by later interventions, we focused on data from the early stages of the epidemic, when there were almost no cases identified outside China. Twelve relevant datasets were identified: (*Ali et al., 2020*; *Ganyani et al., 2020*; *He et al., 2020*; *Liao et al., 2020*; *Li et al., 2020*; *Ren et al., 2021*; *Wu et al., 2020a*; *Xia et al., 2020*; *Yang et al., 2020*; *Zhang et al., 2020*; *Zhanwei et al., 2020*; *Zhao et al., 2020*). In total, the combined dataset contained 2000 pairs, including duplicates. We cross-checked for duplicates in the combined dataset in three steps (see *Appendix 1—figure 1*): First, we removed pairs containing the same 'infector/infectee ID' (leaving 1685 pairs). Second, we looked at datasets containing sex and age information of the contacts and identified as duplicates those with matching sex, age, and symptom-onset date for both cases (identifying 931 unique transmission pairs in these sets). Lastly, we looked at the datasets not containing information regarding the sex and age of the cases (additional 406 pairs) and added to the dataset only pairs with symptom-onset dates that did not occur already in the first group (71 of the 406 cases were added, resulting in 1002 transmission pairs in total). See *Appendix 1—figures 2 and 3* for a visualization of the datasets as a function of the symptom-onset date.

## Statistical model of serial interval data

Following *Klinkenberg and Nishiura, 2011*; *Park et al., 2021*, our model incorporates the possible interaction of the generation interval ($\tau_g$) with the incubation period of the infector ($\tau_i$) using a joint density function, denoted $h(\tau_i, \tau_g)$. The use of a joint distribution allows us to consider a correlation between the two periods. For example, it allows us to assume that infected individuals who develop symptoms later than average are more likely to transmit later than average, given that viral load peaks around the time of symptom onset. This is supported by longitudinal viral load studies showing that viral loads and culture positivity peak around symptoms onset (*Killingley et al., 2022*; *He et al., 2020*) and previous analyses of transmission pairs (*Ferretti et al., 2020a*; *Hart et al., 2021*).

When the epidemic is in equilibrium (i.e., the incidence of infection remains constant over time), we can write down the probability density function $s(\tau \mid \alpha_1, \alpha_2)$ of observing an infector-infectee pair whose symptom-onset dates differ by a specific period (serial interval). This probability density function is conditional on the infection time of the infector $\alpha_1$ and the infectee $\alpha_2$ relative to the symptom-onset time of the infector. As described in *Figure 1*, if we define the symptom onset time of the infector as zero, this means that $\alpha_1 < 0$, and because the infector has to be infected before the infectee, this requires that $\alpha_1 < \alpha_2$. Assuming equilibrium conditions, is equal to the joint distribution describing the generation interval and the incubation period of the infector $h(\tau_i, \tau_g)$, multiplied by the probability density function of the distribution of the infectee's incubation period (denoted $l(\tau - \alpha_2)$. This is a marginal distribution derived from $h$ by integration over $\tau_g$, $l(\tau_i) = \int_0^\infty h(\tau_i, \tau_g) d\tau_g$:

$$s(\tau \mid \alpha_1, \alpha_2) = h(-\alpha_1, \alpha_2 - \alpha_1) \times l(\tau - \alpha_2), \tag{1}$$

where $\tau$ is the serial interval, and $\alpha_1$, $\alpha_2$ are the infection times. The notations are further presented together with the definitions in *Figure 1*. As is shown in *Equation (1)*, the two distributions $(h(\tau_i, \tau_g), l(\tau_{i2}))$ depend on the relative infection times of both the infector and the infectee ($\alpha_1$ and $\alpha_2$). Although the exact time of infection is typically unknown, a possible exposure time window is provided in many cases. To compensate for the lack of information, the model integrates over all possible combinations of infector and infectee exposure times when estimating the parameters of the distribution from the observed serial intervals of the transmission pairs:

$$S(\tau) = \int_{-\infty}^{0} \int_{\alpha_1}^{\tau} h(-\alpha_1, \alpha_2 - \alpha_1) \times l(\tau - \alpha_2) d\alpha_2 d\alpha_1. \tag{2}$$

Most previous analyses of the serial-interval distributions of COVID-19 have relied on this model, which assumes a constant force of infection (i.e., the per capita rate at which susceptible individuals become infected). However, in the beginning of an epidemic, the number of infections (and therefore the force of infection) increases exponentially, creating a specific 'backward' bias. When the force of infection is increasing exponentially, a cohort of infectors that developed symptoms at the same time is more likely to have been infected recently and thus to have shorter incubation periods, on average, than their infectees. Infectors with short incubation periods will also have short generation intervals due to their correlations, meaning that individuals who transmit early after infection are over-represented. It is important to correct for this bias by adding a factor $e^{r\alpha_1}$ (*Park et al., 2021*):

$$\hat{S}(\tau) = \int_{-\infty}^{0} \int_{\alpha_1}^{\tau} e^{r\alpha_1} h(-\alpha_1, \alpha_2 - \alpha_1) \times l(\tau - \alpha_2) d\alpha_2 d\alpha_1. \tag{3}$$

## Incubation period distribution and growth rate assumptions

We used the incubation period distribution provided by a meta-analysis, which reviewed and aggregated 72 studies, as they likely represent best-available estimates for the wild type (*Xin et al., 2021*). In their meta-analysis, *Xin et al., 2021*, found an increase of the incubation period following the introduction of interventions in China, matching the theoretical framework shown above. Their inferred incubation period distribution includes a correction for the growth rate of the early spread, accordingly.

The daily growth rates in the early outbreak period in Wuhan in particular and in the rest of China were estimated by another study (*Tsang et al., 2020*) to be r=0.08 day⁻¹ and r=0.10 day⁻¹, respectively. In our main analysis, we used the growth rate measured for mainland China (r=0.10 day⁻¹), taken as a mean growth rate representing the dynamic of the early outbreak relevant for most of the

transmission pairs. We further present a sensitivity analysis for this parameter (see Results section). We note that daily growth rate estimates of 0.08–0.10 day$^{-1}$ are lower than previous estimates in the range of 0.17–0.3 day$^{-1}$ (*Kamalich et al., 2020a*; *Park et al., 2020*) due to case ascertainment corrections (*Park et al., 2020*). For the functional form of $h$, we used a bivariate lognormal distribution. Parameters for the incubation period were taken from the meta-analysis (*Xin et al., 2021*) leaving three free parameters: the shape and the scale of the lognormal distribution defining the generation-interval univariate distribution, and a correlation parameter (defined as the correlation between the logged incubation period and the logged generation interval). In order to test the sensitivity of our results to the choice of a lognormal distribution, we also considered the alternative form used in *Ferretti et al., 2020a* in *Appendix 1—figure 6*.

## Maximum likelihood inference of the generation-interval distribution

We then chose the parameters $\theta$ that maximize the likelihood of the observed serial intervals $\tau_j^{obs}$ (the maximum likelihood estimate):

$$\hat{\theta} = argmax_{\theta_h} L\left(\tau_j^{obs} \mid \theta_h\right) = argmax_{\theta_h} \sum_j log\left(\hat{S}\left(\tau_j^{obs}\right)\right). \tag{4}$$

Sequential least squares programming method, implemented in Python, was used to maximize the log-likelihood (*Kraft, 1988*; *Virtanen et al., 2020*). We calculated the uncertainties of the estimates using bootstrapping: the dataset was resampled with replacement (100 times for the main analysis and 100 times for sensitivity analyses) and processed via the maximum likelihood framework. In addition, the growth rate ($r$) was sampled from the uncertainty distribution found in a previous study of the early outbreak in China (*Tsang et al., 2020*). We calculated CI based on the 95% quantiles of the bootstrapping results.

## Sensitivity analyses

We conducted three primary sensitivity analyses to investigate potential biases in our approach. First, we tested how our estimate of the unmitigated generation-interval distribution is sensitive to our cutoff date assumption by varying it between January 11 and January 25. We note that using serial-interval data from later dates are generally less reliable as they are affected by mitigation measures, which prevent late transmissions. Second, we considered the possibility that long serial intervals may be caused by omission of intermediate infections in multiple chains of transmission, which in turn would lead to overestimation of the mean serial and generation intervals. Thus, we refit our model after removing long serial intervals from the data (by varying the maximum serial interval between 14 and 24 days). We also considered 'splitting' these intervals into smaller intervals, but decided this was unnecessarily complex, since several choices would need to be made, and the effects would likely be small compared to the effect of the choice of maximum, since the distribution of the resulting split intervals would not differ sharply from that of the remaining observed intervals in most cases. Finally, we considered the possibility that the lack of negative serial intervals in early serial interval data might have been caused by the incorrect determination of the direction of transmission, especially given limited information about pre-symptomatic transmission in the beginning of the pandemic. In other words, infectees who developed symptoms before their infectors may have been incorrectly identified as a primary case. To test for potential biases, we refitted our model after switching the direction of transmission among randomly selected infector-infectee pairs by varying the number of pairs switched (2, 4, 6, or 8 pairs out of 77) and the maximal serial interval for which order switching is allowed (3, 5, or 7 days). For each combination, the analysis was run 30 times with randomly sampled infector-infectee pairs.

Beyond the primary sensitivity analysis, we also performed several supplementary sensitivity analysis. First, we tested other possible sensitivities of the data to biases based on location of infection, or the literature source of the data. To test the sensitivity to infection location, we stratified the dataset by where the infectors were infected (Wuhan vs. outside of Wuhan) as detailed in the supplementary information. To test for sensitivity to any specific literature source, we repeated the analysis while removing one dataset at a time, including all the transmission events that were duplicated also in other datasets (defined by the infector and infectee ID). Second, the effect of the assumed growth rate was assessed by varying it between 0.04 and 0.16 day$^{-1}$ and the effect of the assumed incubation

period distribution assessed by varying its median parameter between 4 and 5.5 days. Furthermore, the effect of inclusion of severe cases was assessed for both the period in focus (prior to January 17) and later dates. Finally, the sensitivity of the results to the choice of the lognormal bivariate distribution model was tested by comparison with another model distribution (given in *Ferretti et al., 2020a*, see supplementary material for full details).

## Estimation of the basic reproduction number

We estimated the basic reproduction number ($R_0$) using the Euler-Lotka equation (*Wallinga and Lipsitch, 2007*):

$$R_0 = \frac{1}{\int_0^\infty e^{-r\tau} g\left(\tau\right) d\tau}, \tag{5}$$

where $g\left(\tau\right)$ is the distribution of the generation interval and $r$ is the growth rate.

## Extrapolation of the unmitigated generation interval of VOCs

Beyond estimating the unmitigated generation interval for the original wild type of SARS-CoV-2, we also extrapolated the unmitigated generation-interval distributions of the alpha, delta, and omicron variants by combining our estimates with previously inferred viral load trajectories (*Hay et al., 2022*; *Kissler et al., 2021*). Kissler et al. estimated exponential growth and clearance (decay) rates of viral load trajectories across 173 participants from the National Basketball Association between November 28, 2020, and August 11, 2021, including individuals infected by alpha and delta variants. Hay et al. extended the analysis to include an additional 204 individuals who were infected by delta or omicron variants. These studies showed that the overall viral shedding time of the new variants was shorter than the non-VOCs, mainly due to a significant reduction of the clearance time – the duration of the period from the peak viral load back to undetectable level of viral load. Following Kissler et al., we assume that the group of non-VOCs represents the original wild-type variant. We assume that differences in clearance durations reflect biological differences in the rate in which the variant infects the host, and therefore base the extrapolation on the ratio of clearance durations: $\kappa = \frac{c_{WT}}{c_{VOC}} < 1$, where $c_{WT}, c_{VOC}$ are the viral trajectories clearance rate of the wild-type variant and VOC. We scaled the infectiousness profile for the VOCs shortening its time course by $\kappa$:

$$h^{VOC}\left(\tau_i, \tau_g\right) = h^{WT}\left(\kappa\tau_i, \kappa\tau_g\right), \tag{6}$$

where $h^{WT}, h^{VOC}$ are the joint bivariate distribution of incubation period and generation interval of the wild-type variant and VOCs. Since the distribution of infectiousness is lognormal, the scaling affects only one of the parameters of the distribution (the median). See supplementary information for full derivation. The resulting unmitigated generation-interval distribution then estimates the unmitigated infectiousness profile of new variants under a counterfactual scenario, in which behavioral and intervention effects remain the same as in the initial pandemic phase.

Although the connection of viral load levels and infectiousness is not well characterized, previously inferred viral load trajectories qualitatively match the shape of the distribution of transmission probability as a function of the TOST, providing support for our approximation (*Figure 6*). This apparent similarity was also demonstrated in previous studies (*He et al., 2020*; *Jones et al., 2021*; *Marc et al., 2021*).

## Acknowledgements

We would like to thank David Champredon and David Earn for valuable feedback on this manuscript. Funding: Ben B and Joyce E Eisenberg Foundation, The Weizmann CoronaVirus Fund (RM), the Israeli Council for Higher Education (CHE) via the Weizmann Data Science Research Center, and by a research grant from the Estate of Tully and Michele Plesser (RS). RM is the Charles and Louise Gartner Professional Chair. Jonathan Dushoff is supported by the Canadian Institutes of Health Research. YMB is an Azrieli Fellow.

## Additional information

### Funding

| Funder | Grant reference number | Author |
|---|---|---|
| Weizmann Institute of Science | The Weizmann CoronaVirus Fund | Ron Milo |
| Weizmann Institute of Science | Weizmann Data Science Research Center and by a research grant from the Estate of Tully and Michele | Ron Sender |
| Canadian Institute for Health Research | | Jonathan Dushoff |
| Ben B. and Joyce E. Eisenberg Foundation | | Ron Milo |

The funders had no role in study design, data collection and interpretation, or the decision to submit the work for publication.

### Author contributions

Ron Sender, Conceptualization, Resources, Data curation, Software, Formal analysis, Validation, Investigation, Methodology, Writing – original draft, Writing – review and editing; Yinon Bar-On, Sang Woo Park, Conceptualization, Resources, Data curation, Formal analysis, Validation, Investigation, Methodology, Writing – original draft, Writing – review and editing; Elad Noor, Conceptualization, Resources, Data curation, Formal analysis, Supervision, Validation, Investigation, Methodology, Writing – original draft, Writing – review and editing; Jonathan Dushoff, Conceptualization, Resources, Data curation, Formal analysis, Supervision, Funding acquisition, Validation, Investigation, Methodology, Writing – original draft, Writing – review and editing; Ron Milo, Conceptualization, Resources, Data curation, Formal analysis, Supervision, Funding acquisition, Validation, Investigation, Methodology, Writing – original draft, Project administration, Writing – review and editing

### Author ORCIDs

Ron Sender (iD) http://orcid.org/0000-0002-1165-9818
Elad Noor (iD) http://orcid.org/0000-0001-8776-4799
Jonathan Dushoff (iD) http://orcid.org/0000-0003-0506-4794
Ron Milo (iD) http://orcid.org/0000-0003-1641-2299

### Decision letter and Author response

Decision letter https://doi.org/10.7554/eLife.79134.sa1
Author response https://doi.org/10.7554/eLife.79134.sa2

## Additional files

### Supplementary files

• Supplementary file 1. The combined dataset of transmission pairs from the early spread of the COVID-19 epidemic in China.

• MDAR checklist

### Data availability

All study data are included in the article, SI appendix, and Dataset S1. All code is available in Jupyter notebooks found in https://gitlab.com/milo-lab-public/the-unmitigated-profile-of-covid-19-infectiousness, (copy archived at swh:1:rev:5e33057809e940d9b3ecd06a389b07611d15b39e).

The following previously published datasets were used:

| Author(s) | Year | Dataset title | Dataset URL | Database and Identifier |
|---|---|---|---|---|
| Ali L | 2020 | PDGLin/COVID19_ EffSerialInterval_NPI: Serial interval of SARS-CoV-2 was shortened over time by non-pharmaceutical interventions | http://doi.org/10. 5281/zenodo.3940300 | Zenodo, 10.5281/ zenodo.3940300 |
| Ganyani T, Kremer C, Chen D, Torneri A, Faes C | 2020 | Estimating the generation interval for COVID-19 | https://github. com/cecilekremer/ COVID19 | github, COVID19 |
| Yang L | 2020 | COVID19_SHIYAN | http://doi.org/10. 5281/zenodo.3898225 | Zenodo, 10.5281/ zenodo.3898225 |
| Due Z | 2020 | Table S5 | https://github.com/ MeyersLabUTexas/ COVID-19 | Table S5, COVID-19 |

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

## Appendix 1

### Extended methods

#### Literature survey for serial-interval data

A literature survey was conducted in order to gather data on serial intervals of transmission events from published and preprint literature. The survey was composed using a 'google scholar' inquiry containing the phrases: 'serial interval' + 'COVID' + 'China'. Twelve relevant datasets were identified: (*Ali et al., 2020*; *Ganyani et al., 2020*; *He et al., 2020*; *Liao et al., 2020*; *Li et al., 2020*; *Ren et al., 2021*; *Wu et al., 2020a*; *Xia et al., 2020*; *Yang et al., 2020*; *Zhang et al., 2020*; *Zhanwei et al., 2020*; *Zhao et al., 2020*).

#### Calculation of the mean serial interval for cohorts of transmission pairs that occurred on the same day

In order to compensate for the scarce data with early dates of infector's onset, we used a simple probabilistic model with Bayesian inference to derive crude estimates of the mean serial interval as a function of the infector symptoms onset date. For each date, the serial intervals of infectors that developed symptoms on that day were assumed to have a Student's t distribution such that the mean, standard deviation, and degrees of freedoms were random variables with the next prior distributions:

- Mean – half-normal distribution with standard deviation of 20 days.
- Standard deviation – half-normal distribution with standard deviation of 10 days.
- Degrees of freedoms – exponential distribution with mean of 30.

A Markov chain Monte Carlo method (implemented in Python pymc3 library) was then used to estimate the mean serial interval and its uncertainty (see Results and *Figure 2B*).

#### Sensitivity analysis to the period of interest

For each of the dates between January 11 and 25, 2020, we extracted the dataset consisting of the transmission pairs with infector onset symptoms up to that date. We rerun the maximum likelihood framework on the extracted datasets. Furthermore, in order to obtain uncertainty estimates, we used a bootstrapping method. In the bootstrapping process we resampled with replacements 100 times and processed via the maximum likelihood framework. In addition, the growth rate ($r$) was sampled from the distribution found by a study of the early outbreak in China (*Tsang et al., 2020*).

#### Sensitivity analysis to the infection location of the infector

The transmission pairs dataset contains data from various cities and provinces in China. The mitigation steps were enacted at different timepoints across China, first in Wuhan and later in other cities and provinces. Previous analysis showed substantial growth rate differences across provinces (*Kamalich et al., 2020b*), but it seems that when corrected for case ascertainment, the observed difference in growth rate between Wuhan and the rest of China is small (0.08 day$^{-1}$ vs. 0.1 day$^{-1}$) (*Tsang et al., 2020*).

In our main analysis, we do not differentiate transmission pairs by location. Thus, spatial effects could affect our results in two ways: via the estimated growth rate or via the period chosen for analysis as an approximation for unmitigated transmission. *Appendix 1—figure 7* shows the sensitivity of the results to a change in the growth rate in the range of 0.04–0.16 day$^{-1}$; estimates for the mean generation interval change in the range of 8.1–9.1 days. Specifically, assuming a growth rate of 0.08 day$^{-1}$ instead of 0.1 day$^{-1}$ has a minimal effect on the main results of the analysis.

We further test how the duration of unmitigated period affects the results of the analysis when the dataset is stratified by the infection location of the infectors and infectee. *Appendix 1—figure 8* compares the mean observed serial intervals when the infector and infectee were infected in or outside of Wuhan. We expect that Wuhan to Wuhan transmissions will be shorter than transmissions from Wuhan to the rest of China, but we do not find significant differences, as the data for transmission pairs from Wuhan is scarce. We also check the sensitivity of the generation-interval-distribution estimates to our choices of the unmitigated period, when it is defined separately for those pairs whose infector has been infected inside or outside Wuhan (shown in *Appendix 1—figure 9*). Our analysis suggests that reasonable changes in the unmitigated period have minor effects on our main estimates (e.g., the median generation interval and the 90% of the distribution). For example, taking

only pairs with an infector that was infected in Wuhan and developed symptoms until January 15 or pairs with an infector that was infected outside of Wuhan and developed symptoms until January 21 leads to a median generation interval of 6.9 days, in comparison to 7.9 days in our main analysis. Both are substantially larger than previous reports (*Ferretti et al., 2020a*; *He et al., 2020*; *Sun et al., 2020*).

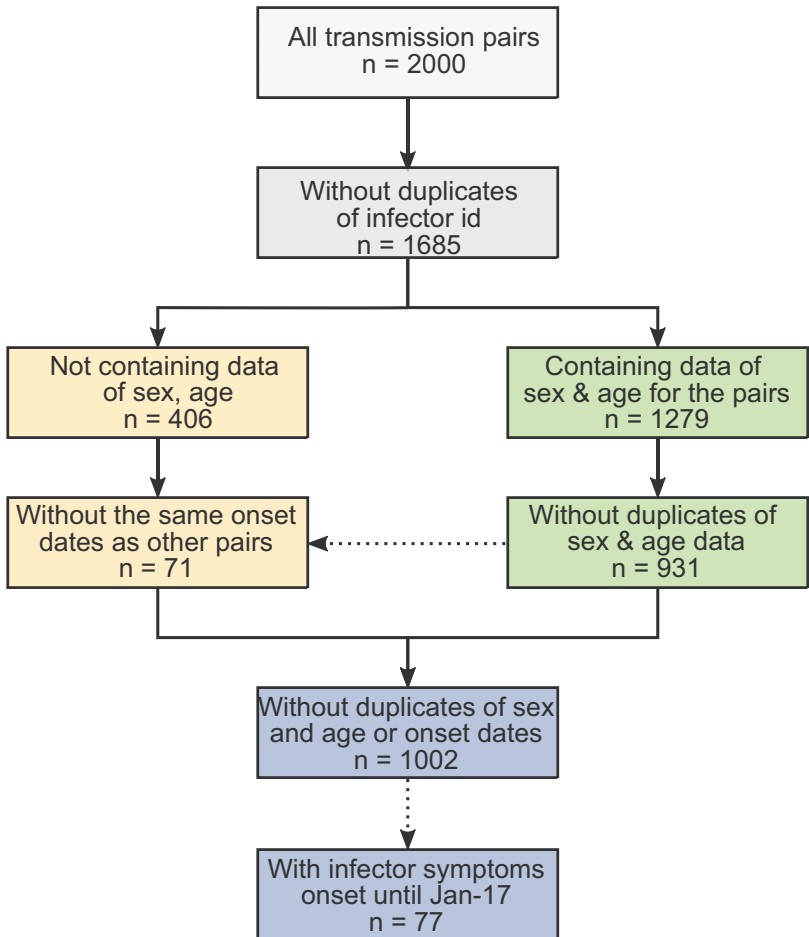

**Appendix 1—figure 1.** The obtained dataset of transmission pairs. The merged datasets were filtered to remove duplicates in three stages: first removal of transmission pairs with the same infector and infectee ID. Second, identification of duplicates sharing the same symptom-onset dates as well as sex and age information. Lastly, transmission pairs without sex and age information were added only if their symptom-onset dates did not already occur in the dataset.

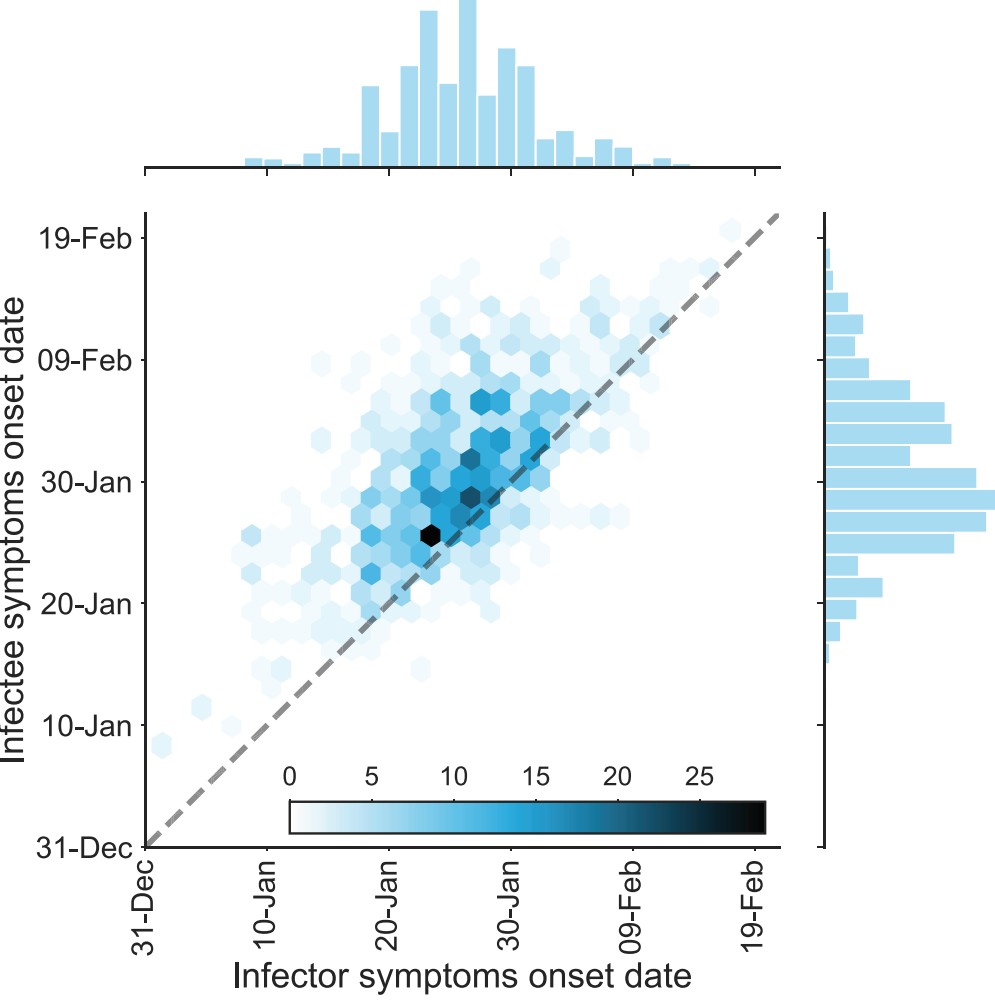

**Appendix 1—figure 2.** The dataset of transmission pairs – infectee symptoms onset date vs. infector symptoms onset date. The observed bivariate distribution of time of symptoms development is shown via a hexbin graph. The plane is divided into hexagons that are colored according to the number of data points in the dataset they represent. The marginal empirical distributions are shown using a histogram on the sides. The dotted line represents data points for which the symptoms' onset date of the infector is the same as that of the infectee.

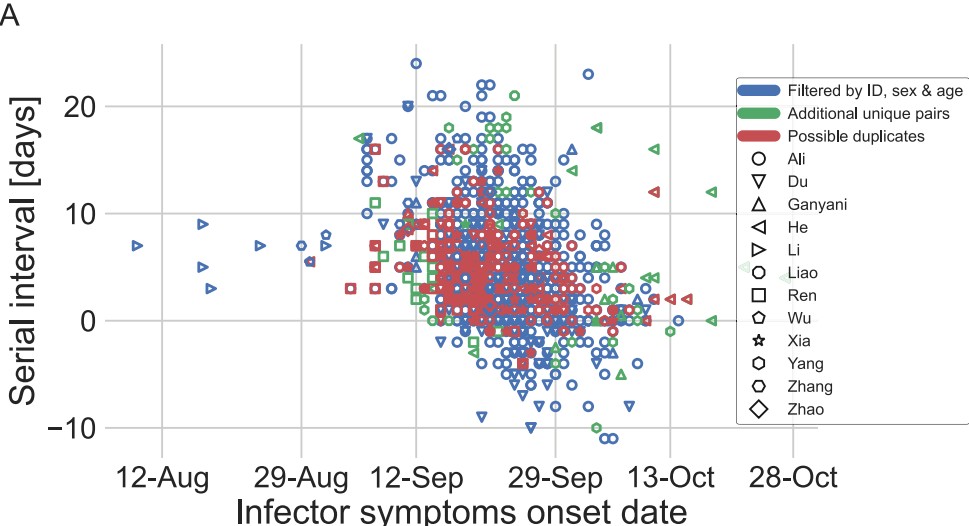

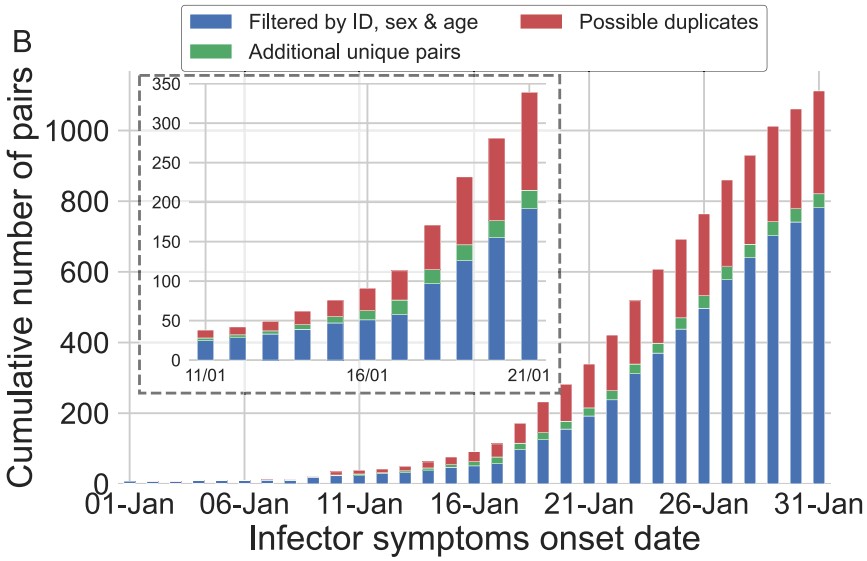

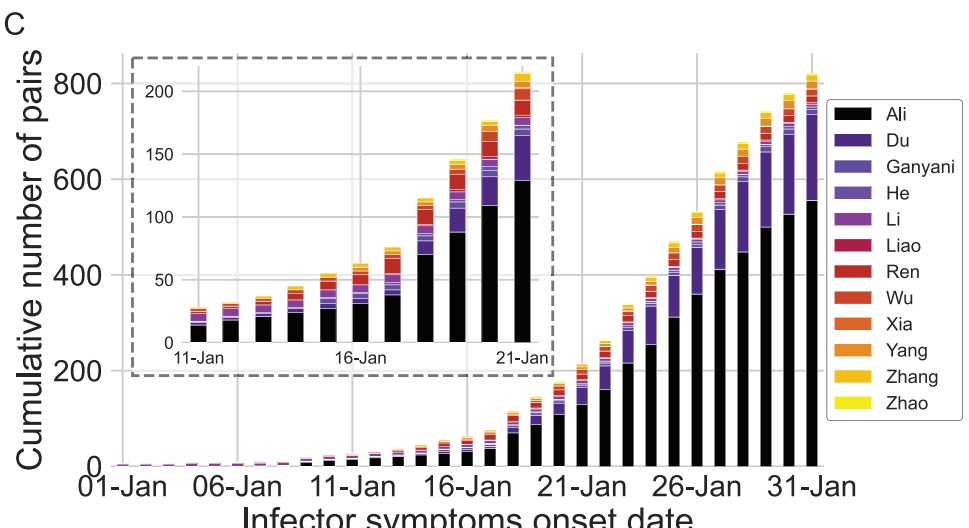

**Appendix 1—figure 3.** The dataset of transmission pairs as a function of the infector symptoms onset date. (A) Scatter plot of serial intervals plotted against the symptoms onset date of the infector. The three levels of filtering are color-coded, while the shape of the marker represents the reference from which the data was taken. (B) The cumulative number of cases as a function of the infector symptoms onset date, where the data is divided between the three levels of filtration. The inset focuses on the period that at its end interventions were made. (C) The cumulative number of cases as a function of the infector symptoms onset date, where the data is divided between the data sources. The dataset is shown after filtrating by ID, sex, and age and the addition of unique pairs (no duplicates). The inset focuses on the period that at its end interventions were made.

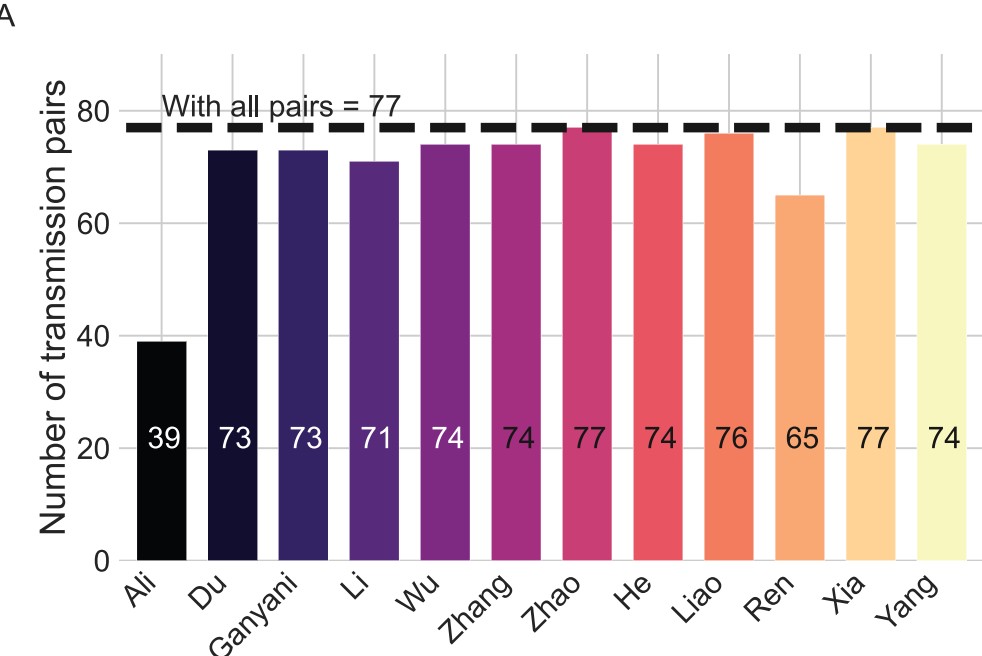

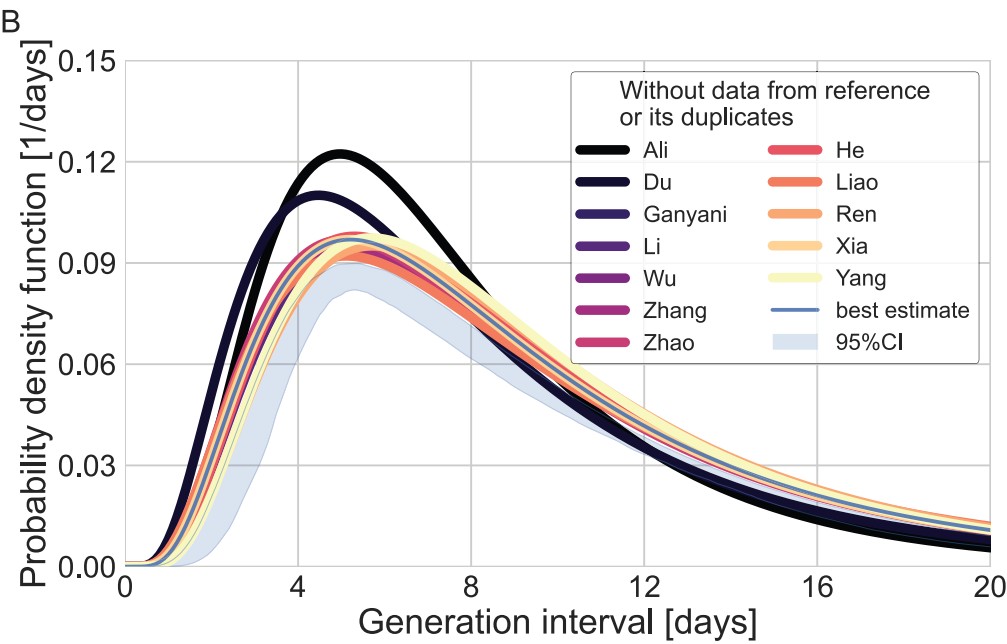

**Appendix 1—figure 4.** Sensitivity analysis regarding the inclusion of a dataset from a specific source. Beginning with the complete dataset of transmission pairs with infector onset until January 17, 2020 (after filtering by ID, sex, and age and adding unique pairs, see *Appendix 1—figure 1*), partial datasets were created by omitting all transmission pairs from each source of data and all its duplicates in the other datasets. (**A**) The number of transmission pairs for each of the partial datasets, excluding pairs from a specific source dataset and their duplicates in other datasets. (**B**) Maximum likelihood estimates of the bivariate incubation period and generation-interval distribution. The uncertainty range of the maximum likelihood estimate of the complete dataset is also shown for comparison.

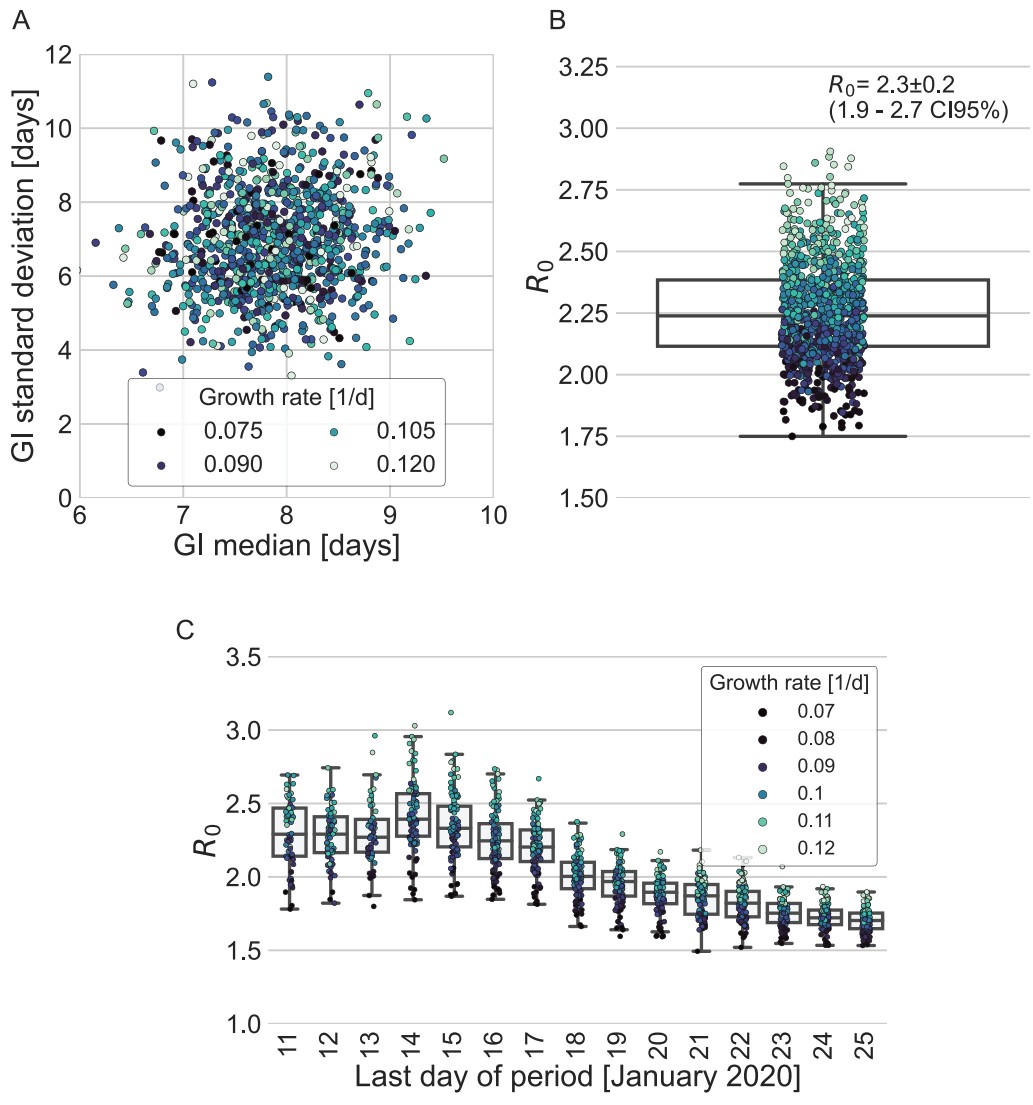

**Appendix 1—figure 5.** Estimates of $R_0$ based on the inferred generation-interval distribution. (**A–B**) Bootstrapping results of the parameters of the generation-interval distribution and the resulting estimates for $R_0$. In the process of bootstrapping, the dataset of 77 transmission pairs was resampled with returns. In addition, the growth rate (*r*) was sampled from the distribution found in a recent study (***Tsang et al., 2020***). (**A**) Estimates of the mean and standard deviation of the generation interval. Each point represents the maximum likelihood estimate for a single run in a bootstrap process. The point was colored according to the sampled growth rate. (**B**) The distribution of estimates of $R_0$ derived from the generation-interval distribution and growth rate. The box represents the interquartile range (percentiles 25–75) and the whiskers represent the maximal range of the distribution apart from outliers (defined as data points exceeding the interquartile range by a factor of 1.5). The mean (with its 95% confidence interval) and the standard deviation is given in the legend. The points are colored according to the sample growth rate, as in panel A. (**C**) The dependence of $R_0$ estimates on the period taken in the analysis. The boxes represent the interquartile range (percentiles 25–75) and the whiskers represent the maximal range of the distribution apart from outliers (defined as data points exceeding the interquartile range by a factor of 1.5). The points are colored according to the sample growth rate, as described in the legend.

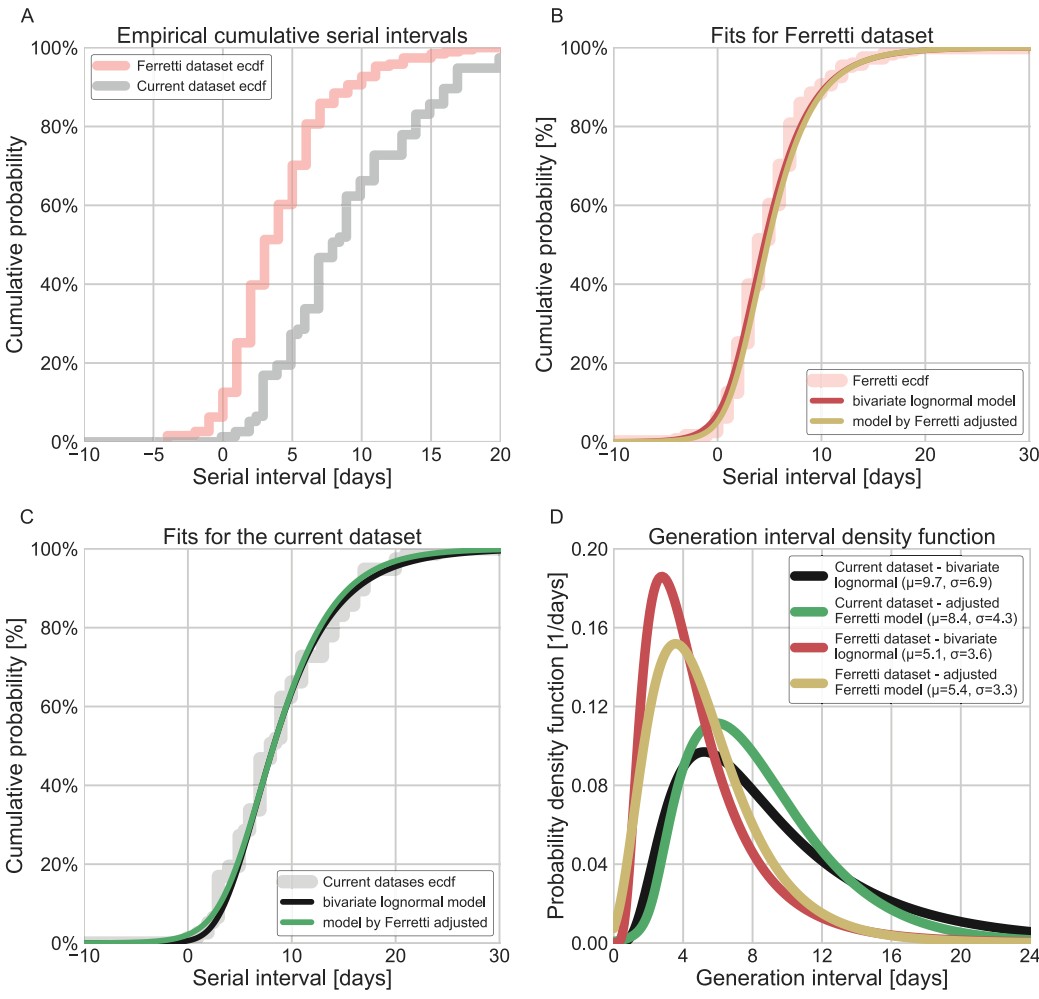

**Appendix 1—figure 6.** Comparison of the current dataset and model with that of *Ferretti et al., 2020a*. The maximum likelihood framework was used to fit both the current dataset and the one provided in the Figure 1 of Ferretti et al. The datasets were fit using either the lognormal bivariate model described in the Methods section, or a reconstructed model following *Ferretti et al., 2020a*, adjusted by adding a parameter for shifting the time from onset of symptoms to transmission (TOST) function over the x-axis. (**A**) The empirical cumulative distribution of serial intervals, comparison between the dataset of *Ferretti et al., 2020a*, and the current dataset curated in this study. (**B**) Maximum likelihood fits for the dataset provided in Figure 1 of Ferretti et al. (**C**) Maximum likelihood fits for the current dataset. (**D**) The marginal generation-interval distributions of the maximum likelihood fits. The mean and standard deviation are provided in the legend.

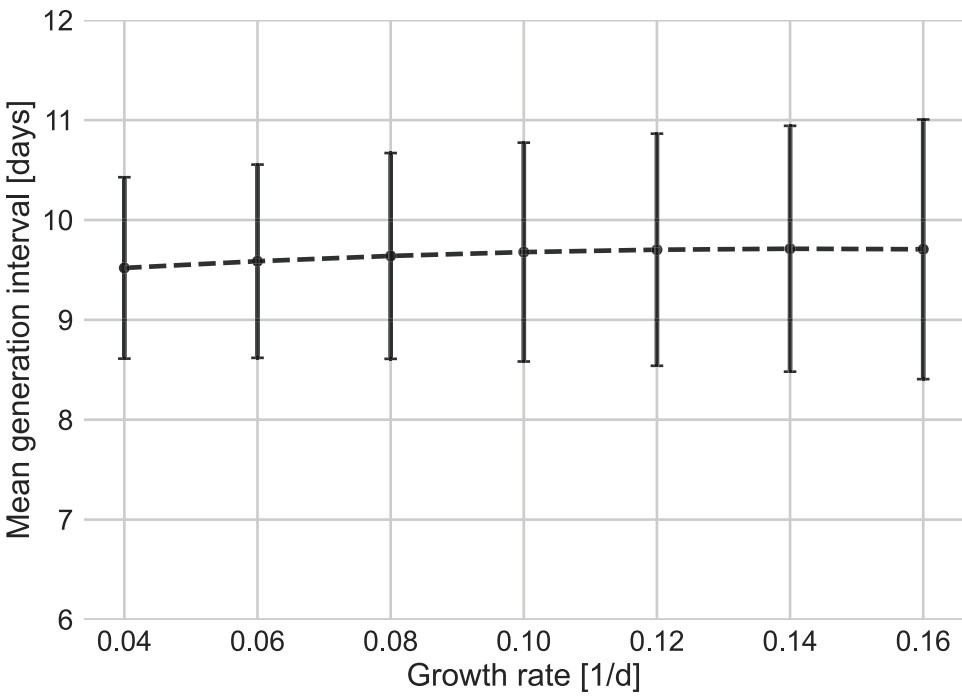

**Appendix 1—figure 7.** Sensitivity analysis to the growth rate. The mean of the generation-interval distributions were estimated using the maximum likelihood fits for the dataset with growth rates in the range of 0.04–0.16 day$^{-1}$. Estimates of the uncertainty were obtained using bootstrapping (30 runs for each value of r). Error bars represent 95% confidence interval.

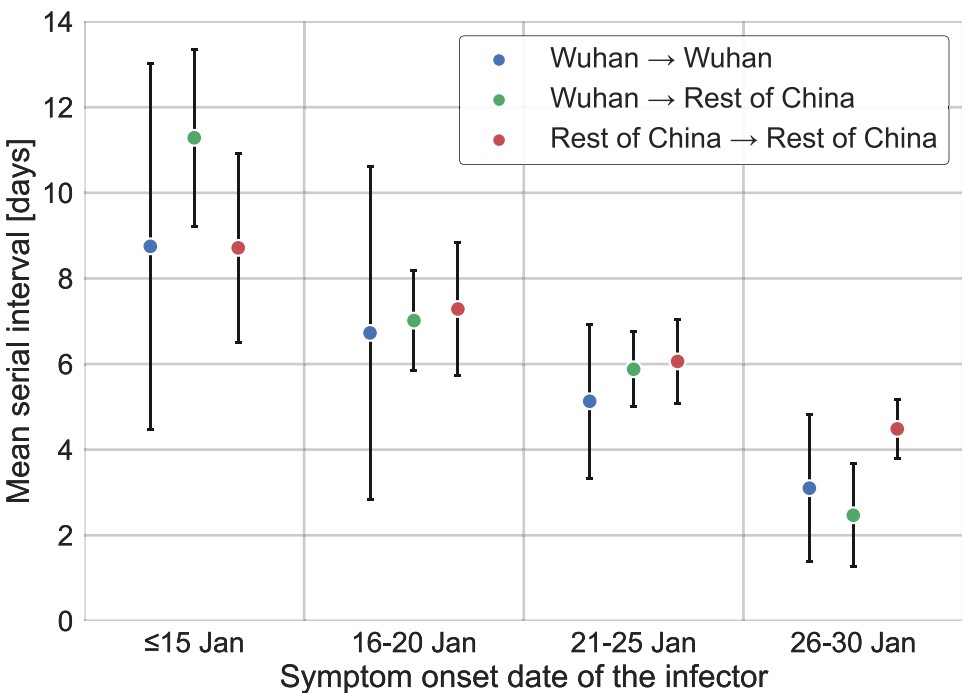

**Appendix 1—figure 8.** Stratification of the serial interval data by the location of infection. A comparison of the mean of the observed distribution of serial intervals divided to four time periods of the infector symptom onset, and stratified by the infection location of the infector and infectee. Error bars represent 95% confidence interval.

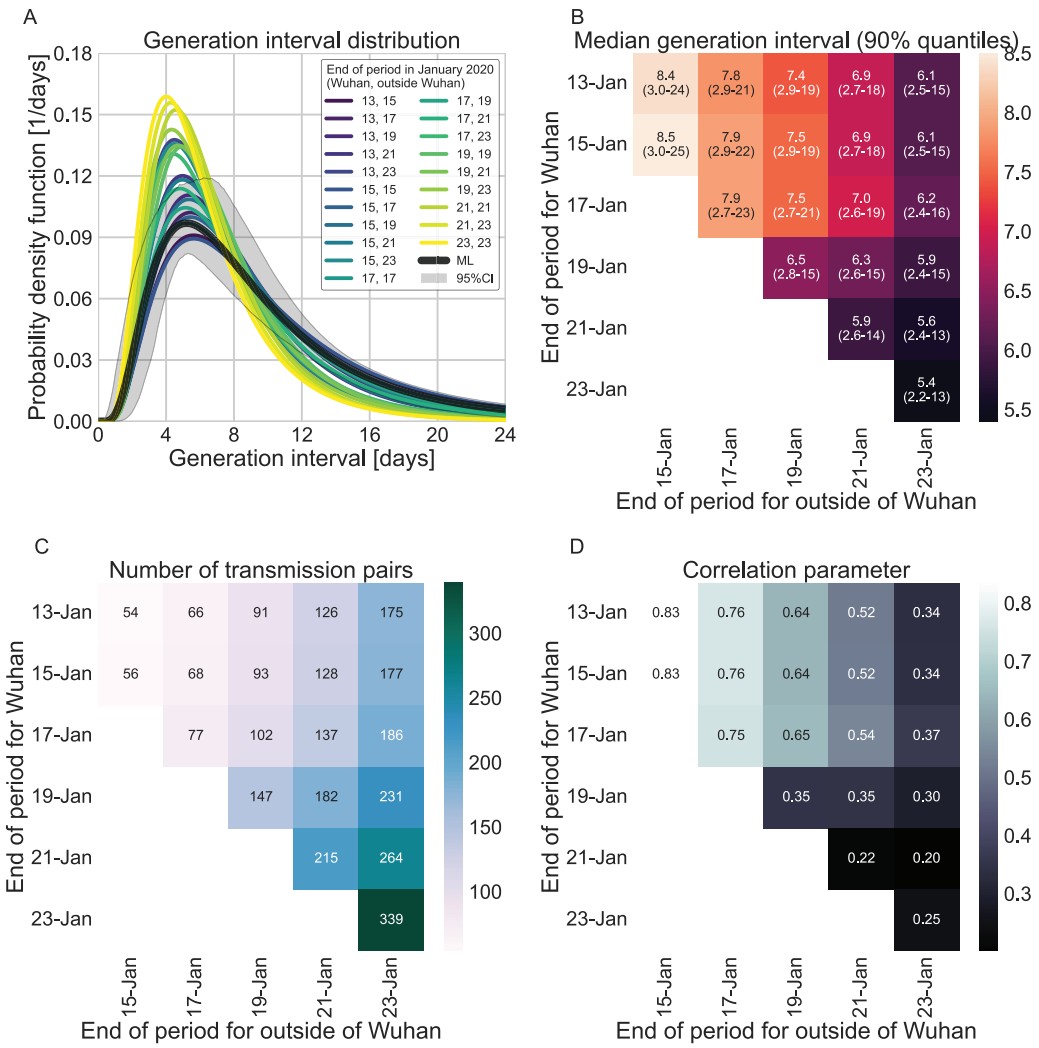

**Appendix 1—figure 9.** Sensitivity analysis to the definition of the period of interest for infectors that were infected in or outside Wuhan.

A comparison of the resulting maximum likelihood estimates where the period of interest was defined separately for infectors who were infected in or outside of Wuhan. (A) The shapes of the resulting generation-interval distribution. For comparison, the main analysis' maximum likelihood is presented together with its 95% interval (the highlighted area). (B) Estimates of the median generation intervals and the 90% interquartile range as function of the period of interest, defined separately for infectors who were infected in or outside of Wuhan. (C) Number of transmission pairs analyzed as function of the period of interest, defined separately for infectors who were infected in or outside of Wuhan. (D) Estimates for the correlation between incubation period and generation-interval parameter as function of the period of interest, defined separately for infectors who were infected in or outside of Wuhan.

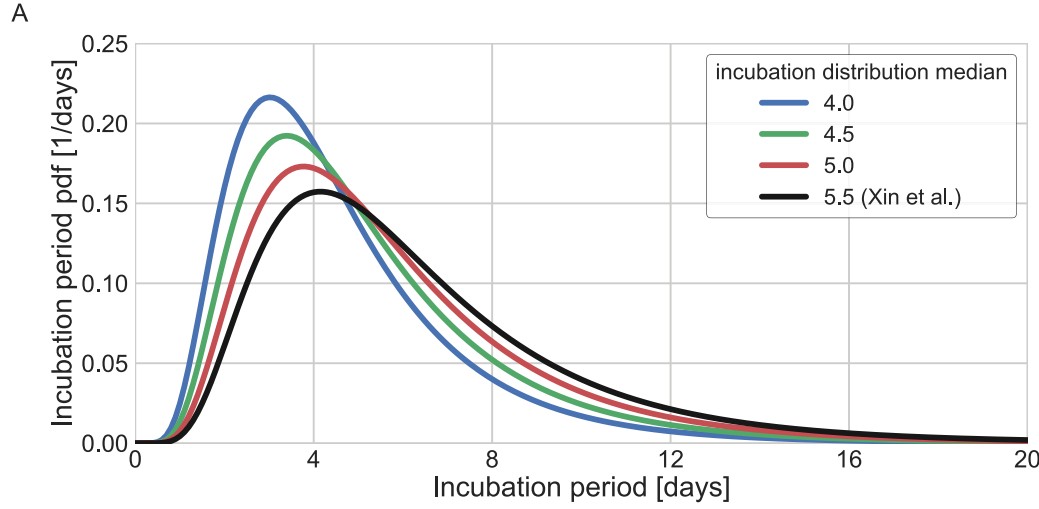

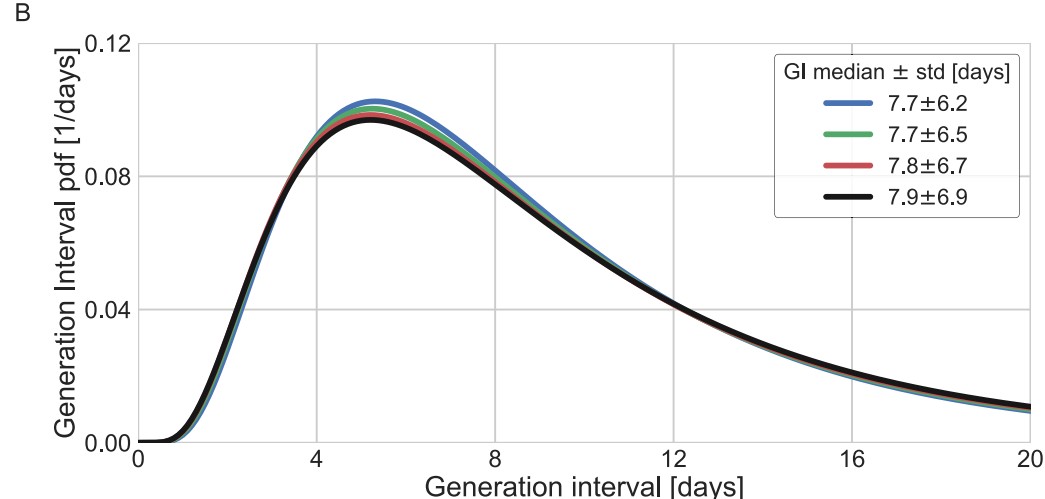

**Appendix 1—figure 10.** Sensitivity analysis to the assumed incubation period distribution. The generation-interval distribution is inferred by maximum likelihood when the incubation period distribution is assumed to have a different median (scale parameter). (A) The probability density functions of the incubation distributions taken in the sensitivity analysis, corresponding medians of 4, 4.5, 5, and 5.5 days. (B) The resulting probability density function of the generation intervals. Legend reports the median and standard deviation of each of the distributions.

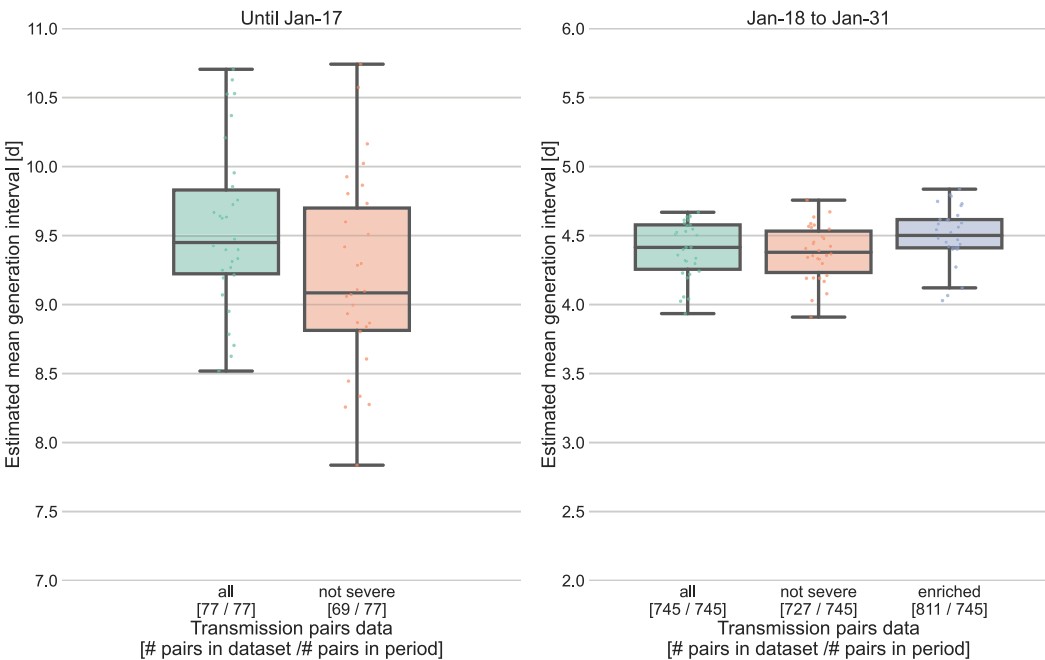

**Appendix 1—figure 11.** Sensitivity analysis for the severity of cases. Severe cases (including death) are over-represented in the period prior to January 17, with 8 out of 77 cases, compared to 18 out of 745 in the period of January 18–31. The effect of inclusion of severe cases was analyzed by comparing the means of the estimated generation-interval distribution, separately for the two periods in question, using the inference framework with 30 bootstrapping runs. For the earlier period, the estimated mean were compared for the dataset with or without the severe cases. For the later period, we also compared the results to an enriched dataset in which the severe cases were oversampled (using bootstrapping) such that the proportion of severe cases matches that during the earlier period (10%). The boxes represent the interquartile range (percentiles 25–75) and the whiskers represent the maximal range of the distribution apart from outliers (defined as data points exceeding the interquartile range by a factor of 1.5).

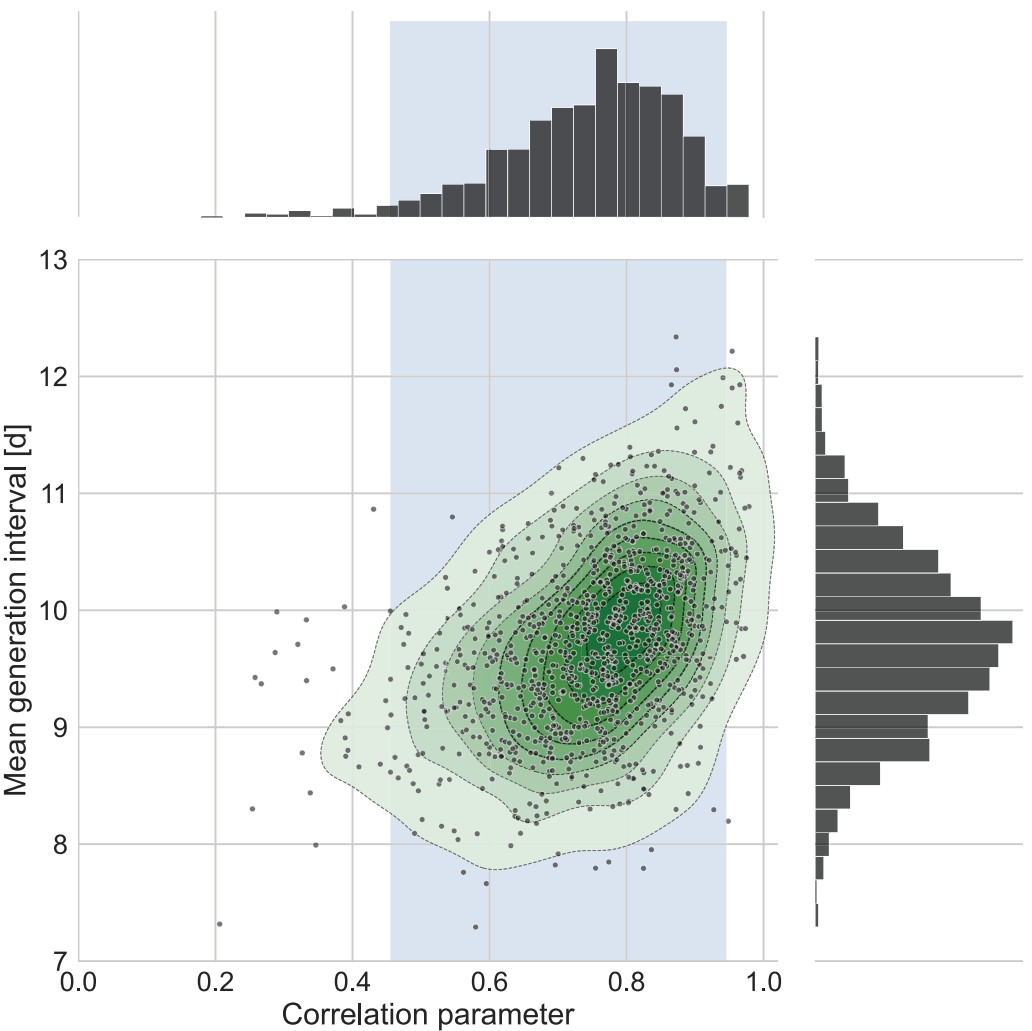

**Appendix 1—figure 12.** The connection between the estimated mean generation interval and the correlation parameter. One thousand bootstrap runs were conducted for estimation of the uncertainty of the inferred bivariate distribution of incubation period and generation interval. The mean of the generation-interval distribution was derived and plotted against the estimated correlation parameter for each run together with contours representing quantiles of equal probability (central panel). The margin distributions of the two parameters are shown on the right and upper panel. The shaded region corresponds to the 95% confidence interval of the correlation parameter (0.45–0.95).

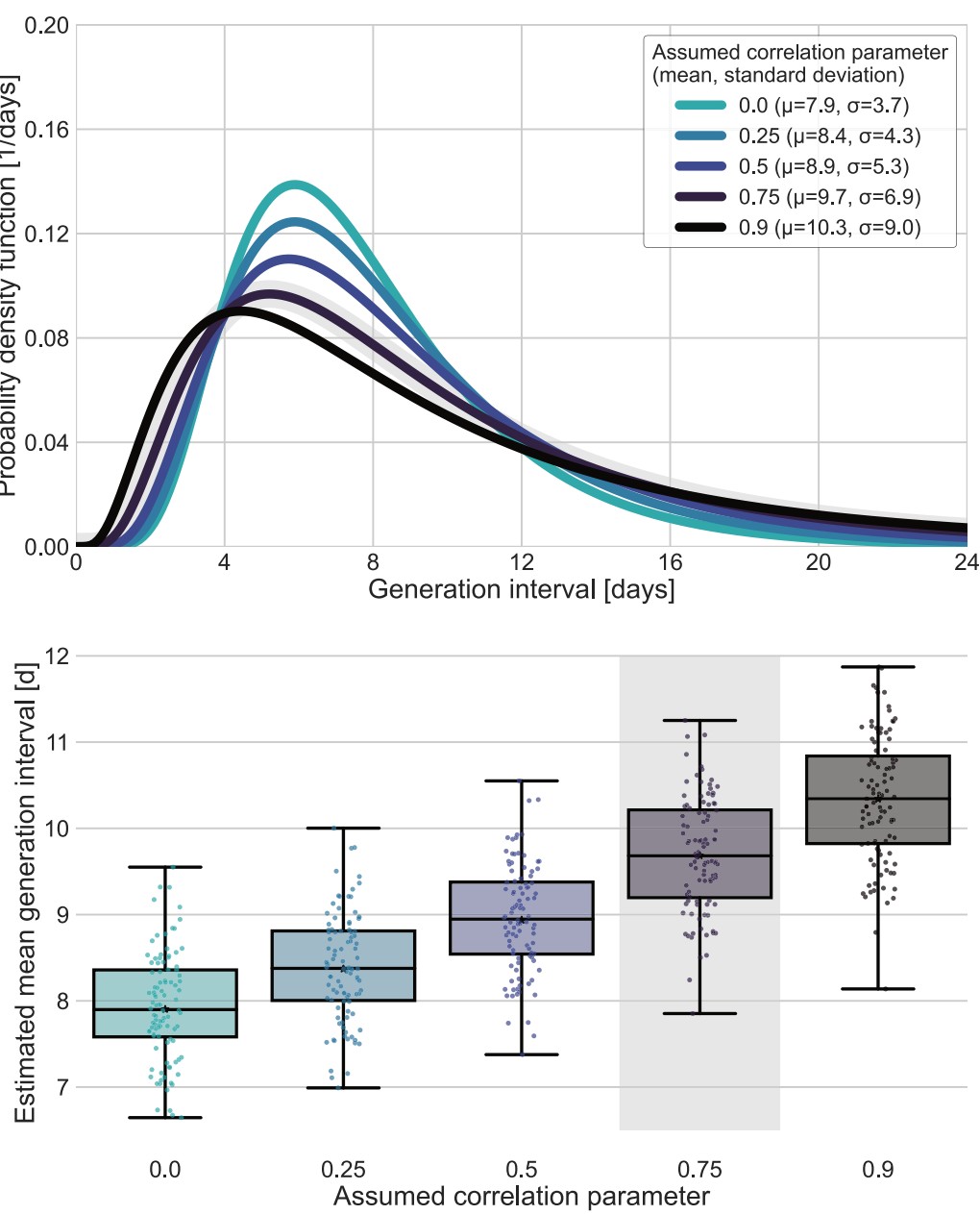

**Appendix 1—figure 13.** Sensitivity analysis for fixed correlation parameters. Maximum likelihood estimates of the bivariate incubation period and generation-interval distribution were made for datasets containing the transmission pairs with infector onset date up January 17 while the correlation parameter was fixed at one of 0, 0.25, 0.5, 0.75, 0.9. Best estimates were derived for each of the datasets. Uncertainty estimates were derived by bootstrapping, through sampling with replacement from the dataset and sampling from the distribution of growth rates (*Tsang et al., 2020*) N=100 times. (**A**) Best estimate for the generation-interval distribution probability density function for assumed correlation parameters. (**B**) Best estimates and distributions of the mean generation interval. A black star marks best estimates. Ranges are given as boxplots. The box represents the interquartile range (percentiles 25–75) and the whiskers represent the maximal range of the distribution apart from outliers (defined as data points exceeding the interquartile range by a factor of 1.5). The fitted correlation (0.75) is highlighted in gray shade.

## Comparison with another model of infectiousness

To check whether these results are sensitive to our choice of using a bivariate lognormal distribution to characterize the joint distribution of the generation interval and the incubation period, we repeated our analysis using a different functional form using an adjusted logistic TOST model following *Ferretti et al., 2020a*. Ferretti et al. modeled the transmission by assuming a TOST distribution with a skewed-logistic shape that is dependent on the incubation period (only on the left side).

$$\text{P}_{tost}\left(t \mid t_i\right) \propto \begin{cases} \dfrac{e^{-(t+m_r)/\sigma}}{\left(e^{-(t+m_r)/\sigma}\right)^{\alpha+1}} & \text{if } t \geq 0 \\[2em] \dfrac{e^{-(t+m_l)/(\sigma t_i/\tau)}}{\left(e^{-(t+m_l)/(\sigma t_i/\tau)}\right)^{\alpha+1}} & \text{if } t < 0 \end{cases}$$

While $t_i$ is the specific incubation period, $\tau$ is the mean incubation period (5.42 days), $\alpha$, $\sigma$ are parameters determining the shape of the distribution, and $m_l$, $m_r$ are the median of the distribution of the negative and positive sides ($m_l = \frac{\sigma t_i}{\tau} ln\left(2^{\frac{1}{\alpha}}-1\right)$, $m_r = \sigma ln\left(2^{\frac{1}{\alpha}}-1\right)$).

We adjusted their model by an additional parameter ($l$) enabling shifting the TOST along the time axis.

$$\text{P}_{tost}\left(t \mid t_i\right) \propto \begin{cases} \dfrac{e^{-(t+m_r-l)/\sigma}}{\left(e^{-(t+m_r-l)/\sigma}\right)^{\alpha+1}} & \text{if } t \geq 0 \\[2em] \dfrac{e^{-(t+m_l-l)/(\sigma t_i/\tau)}}{\left(e^{-(t+m_l-l)/(\sigma t_i/\tau)}\right)^{\alpha+1}} & \text{if } t < 0 \end{cases}$$

We use our maximum likelihood framework to estimate the generation-interval distribution of the adjusted form based on our compiled dataset. Furthermore, we fitted both our models and the adjusted model to the serial-interval dataset provided in Ferretti et al. supplementary Figure S1. Results of the comparison are presented in *Appendix 1—figure 6*.

## Extrapolation of the unmitigated generation interval of VOCs

Beyond estimating the unmitigated generation interval for the original wild type of SARS-CoV-2, we also extrapolated the unmitigated generation-interval distributions of the alpha and delta and omicron variants by combining our estimates with previously inferred viral load trajectories (*Hay et al., 2022*; *Kissler et al., 2021*). Following Kissler et al. notation, we use the group of non-VOCs as representing the original wild-type variant. As discussed in the Methods section, we assume that differences in clearance durations reflect biological differences in the rate in which the variant infects the host, and therefore base the extrapolation on the ratio of clearance durations: $\kappa = \frac{c_{WT}}{c_{VOC}} < 1$, where $c_{WT}, c_{VOC}$ are the viral trajectories clearance rate of the wild-type variant and VOC.

We scaled the infectiousness profile for the VOCs shortening its time course by:

$$h^{VOC}\left(\tau_i, \tau_g\right) = h^{WT}\left(\kappa\tau_i, \kappa\tau_g\right)$$

where $h^{WT}, h^{VOC}$ are the joint bivariate distribution of incubation period and generation interval of the wild-type variant and VOC, respectively. Following the use of a bivariate lognormal distribution for $h$, we get a simple expression for the extrapolated generation-interval distribution:

$$g^{VOC}\left(\tau_g\right) = \int_0^\infty h^{VOC}\left(\tau_i, \tau_g\right) d\tau_i = \int_0^\infty h^{WT}\left(\kappa\tau_i, \kappa\tau_g\right) d\tau_i = g^{WT}\left(\kappa\tau_g\right),$$

where $g^{WT}, g^{VOC}$ are the generation-interval distribution of the variants.

Therefore, $g^{voc}(\tau_g) \sim lognormal(\kappa\mu, \sigma)$
where $\mu, \sigma$ are the scale and shape parameters of $g^{WT}$ such that $g^{WT}\left(\tau_g\right) \sim lognormal\left(\mu, \sigma\right)$.

Thus, the scaling was achieved by multiplying the log mean parameter of the lognormal incubation period and generation-interval distributions by the ratios of the clearance durations. This scaling affects only one of the parameters of the distribution (the median), keeping the squared coefficient of the variation constant.

