## [Editor Report]

By analyzing a carefully curated dataset of cases observed early, and adjusting for multiple forms of bias, this study provides convincing evidence that in the absence of public health interventions, the duration of infectiousness of COVID-19 (original variant) is longer than previously estimated. These important findings improve our ability to model counterfactual intervention-free scenarios, add to evidence that interventions have reduced the duration of infectiousness, and provide an example of how to navigate the biases and pitfalls inevitably present in outbreak data.

---

## [Decision Letter]

**Decision letter after peer review:**

Thank you for submitting your article "The unmitigated profile of COVID-19 infectiousness" for consideration by *eLife*. Your article has been reviewed by 3 peer reviewers, one of whom is a member of our Board of Reviewing Editors, and the evaluation has been overseen by Aleksandra Walczak as the Senior Editor. The following individual involved in the review of your submission has agreed to reveal their identity: James A Hay (Reviewer #3).

Essential revisions:

(1) All three reviewers raised questions about the potential impact of ascertainment bias and small sample size in the unmitigated transmission pair data. Please address potential impacts on the results, and qualify the conclusions if appropriate.

(2) Address questions from two reviewers about the accuracy of fixed incubation period estimates obtained from a meta-analysis. Should these be corrected for the same biases that affect generation interval estimates?

(3) Please provide more detail about the methods used to estimate R0 and the generation interval of variants of concern. Please also consider editing the methods for clarity and readability by a general audience.

(4) In order to make the manuscript more accessible to a general audience, please provide a clearer explanation of why short forward intervals are overrepresented in a growing epidemic. Consider including a diagram or simulation, as suggested by Reviewer 3.

(5) Please address the impact of uncertainty in viral load trajectories on individual generation times, on the residual fraction, and on our ability to infer generation intervals for variants of concern using viral load trajectories. On a related note, please consider modifications to Figure 6a so that it is easier to visualize whether the viral load trajectory aligns well with the claim that 18% of transmission occurs >14d after infection.

*Reviewer #1 (Recommendations for the authors):*

1. The methods section is complete, but it might be easier to follow with more attention to organization, transitions, and maybe with additional subheadings. In particular, it would be helpful if key details like which parameters are being estimated, and which data you're fitting to, were easier to locate in this section.

2. The Introduction and Methods cover a lot of ground summarizing all the forms of bias and adjustment that go into producing an accurate unmitigated estimate, and it is currently a bit hard to keep track of all these details. It could be helpful to provide some sort of list, table, or summary paragraph to help readers keep track of all the forms of bias and adjustment that this analysis deals with, including references where appropriate. It would also be helpful to more clearly state that the main contribution of this study is to collect and apply all these statistical corrections to a carefully curated dataset.

3. I got tripped up by this statement on page 12:

"We find that our framework is able to properly reproduce the realized serial interval distribution given the growth rate in the early stages of the outbreak in Wuhan, China (Figure 3b)."

Aren't the models fit to the SI data-meaning that we expect this result and should be alarmed by anything else? I think that this is just a wording issue and that what you're trying to say here is something like, "With or without the growth-rate adjustment, the model was able to fit the observed serial interval data well (Figure 3b)". But with the current phrasing, it sounds (at least to me) like this is being presented as some sort of independent validation of the model. For the same reason, I'd consider changing "estimated SI" to "fitted SI' in the Figure 3b legend.

*Reviewer #2 (Recommendations for the authors):*

As well as the broad comments made in the public review, I had the following comments:

– "This dataset includes a total of 77 transmission pairs with a mean serial interval of 9.1 days (7.9-10.2 95% CIs) and a standard deviation of 5.2 days. This is substantially longer than the mean of 7.8 days suggested by Ali et al" – is it possible to quantify this difference statistically (e.g. with a test for difference in means between the samples)? Given a mean of 9.1 days and SD=5.2, it wouldn't seem implausible for a random subsample from this dataset to have a mean of 7.8?

– Could the authors clarify which formula they used from Wallinga and Lipsitch (2007) to calculate R0 from generation time, as the exact calculation will depend on assumptions about the distribution of generations etc? I presume the authors used an appropriate formulation but would be useful to state explicitly. The finding that the early R0 is similar despite a longer generation time seems a bit counter-intuitive, so it would be helpful to have some more discussion about what's happening here.

– It would be useful to give some intuition about why changing the baseline incubation period had a limited effect on the results. Is this because the epidemic phase adjustment dominates in the calculation?

– The methods for scaling the generation interval for other VOCs are described briefly in the caption to Figure 6, but it would be helpful to have the calculation given explicitly in the methods, so there is no ambiguity in terms like "ratio of the clearance's durations". Also in this figure, it's unclear where Α line is in B and C, so worth mentioning in the caption. Finally, I didn't follow this sentence: "The inset shows a zoom-in on the period of 12-24 days after exposure, a period in which there is a substantial difference between the current estimate and those from previous studies." Are the presented estimates not all new ones derived from the current study and viral shedding data.

– I appreciate that not all of these studies were available at the time of submission, but it could be helpful to update the discussion to also place the results in the context of more recent viral culture duration from serial swabbing data (Chu et al., JAMA Int Med 2022) and/or shedding profiles in human challenge data (Killingley et al., Nature Med 2022).

---

## [Author Response]

Essential revisions:(1) All three reviewers raised questions about the potential impact of ascertainment bias and small sample size in the unmitigated transmission pair data. Please address potential impacts on the results, and qualify the conclusions if appropriate.

We recognize the potential biases in the transmission pairs data. We therefore developed an extensive framework of sensitivity analyses for identifying biases that could substantially affect the results. In the Results section and figure 5, we show that the main study result, that the unmitigated generation-interval distribution is longer than previously estimated, is robust to reasonable amounts of ascertainment bias. We discuss this point at length and have added several supplemental figures to support this claim.

As reviewer #3 mentioned, we conducted a sensitivity analysis for the inclusion of the longest serial intervals, to investigate possible effects of missing links in the longest transmission pairs. We also discuss why we think it’s not necessary to explicitly model the short intervals that may be unobserved due to missing links.

“Second, we considered the possibility that long serial intervals may be caused by omission of intermediate infections in multiple chains of transmission, which in turn would lead to overestimation of the mean serial and generation intervals. Thus, we refit our model after removing long serial intervals from the data (by varying the maximum serial interval between 14 and 24 days). We also considered “splitting” these intervals into smaller intervals, but decided this was unnecessarily complex, since several choices would need to be made, and the effects would likely be small compared to the effect of the choice of maximum, since the distribution of the resulting split intervals would not differ sharply from that of the remaining observed intervals in most cases.”

We added to the discussion text regarding the effect of possible bias in the dataset, explicitly specifying the ascertainment bias.

“Our analysis relies on datasets of transmission pairs gathered from previously published studies and thus has several limitations that are difficult to correct for. Transmission pairs data can be prone to incorrect identification of transmission pairs, including the direction of transmission. In particular, presymptomatic transmission can cause infectors to report symptoms after their infectees, making it difficult to identify who infected whom. Data from the early outbreak might also be sensitive to ascertainment and reporting biases which could lead to missing links in transmission pairs, causing serial intervals to appear longer (For example, people who transmit asymptomatically might not be identified). Moreover, when multiple potential infectors are present, an individual who developed symptoms close to when the infectee became infected is more likely to be identified as the infector. These biases might increase the estimated correlation of the incubation period and the period of infectiousness. We have tried to account for these biases by using a bootstrapping approach, in which some data points are omitted in each bootstrap sample. The relatively narrow ranges of uncertainty suggest that the results are not very sensitive to specific transmission pairs data points being included in the analysis. We also performed a sensitivity analysis to address several potential biases such as the duration of the unmitigated transmission period, the inclusion of long serial intervals in the dataset, and the incorrect ordering of transmission pairs (see Methods). The sensitivity analysis shows that although these biases could decrease the inferred mean generation interval, our main conclusions about the long unmitigated generation intervals (high median length and substantial residual transmission after 14 days) remained robust (Figure 5).”

(2) Address questions from two reviewers about the accuracy of fixed incubation period estimates obtained from a meta-analysis. Should these be corrected for the same biases that affect generation interval estimates?

In our analysis we use the incubation period distribution from Xin et al. 2021 which already considers the backward bias caused by the expanding epidemic with the corrected growth rate of 0.1/d. Xin et al. showed in their meta-analysis that the mean incubation period reported by the various sources changed according to the dates used by the source. Incubation periods prior to the peak of the epidemic in China were lower than ones from after the peak, in a manner that coincided with the backward correction they performed (using a similar derivation to that suggested by Park et al. 2021). Accordingly, the distribution of incubation period they report is the intrinsic incubation period, after correction for the growth rate of the initial spread in China. We added two sentences in our methods section to clarify this point:

“In their meta-analysis, Xin et al. found an increase of the incubation period following the introduction of interventions in China, matching the theoretical framework shown above. Their inferred incubation period distribution includes a correction for the growth rate of the early spread, accordingly.”

Furthermore, we perform a sensitivity analysis for the shape of the incubation period distribution, and show that it has a minor effect on our conclusions (Appendix 1—figure 10).

(3) Please provide more detail about the methods used to estimate R0 and the generation interval of variants of concern. Please also consider editing the methods for clarity and readability by a general audience.

We made some edits to the methods section in order to make it more accessible and clear, for example, we added subheadings for the various sections, added a section explaining the derivation of the basic reproduction number, and clarified the section regarding the VOCs extrapolations:

“We estimated the basic reproduction number (R0) using the Euler-Lotka equation (Wallinga and Lipsitch 2007):, (5)R0=1∫0∞e−rτg(τ)dτ

where g(τ) <milestone-start /> <milestone-end /> is the distribution of the generation interval and r  is the growth rate.”

“Beyond estimating the unmitigated generation interval for the original wild type of SARS-CoV-2, we also extrapolated the unmitigated generation-interval distributions of the α, δ and omicron variants by combining our estimates with previously inferred viral load trajectories (Kissler et al. 2021; Hay et al. 2022). Kissler et al. estimated exponential growth and clearance (decay) rates of viral load trajectories across 173 participants from the National Basketball Association between November 28, 2020, and August 11, 2021, including individuals infected by α and δ variants. Hay et al. extended the analysis to include an additional 204 individuals who were infected by δ or omicron variants. These studies showed that the overall viral shedding time of the new variants was shorter than the Non-VOC variants, mainly due to a significant reduction of the clearance time – the duration of the period from the peak viral load back to undetectable level of viral load. Following Kissler et al., we assume that the group of non-VOC variants represents the original wild type variant. We assume that differences in clearance durations reflect biological differences in the rate in which the variant infects the host, and therefore base the extrapolation on the ratio of clearance durations: κ=cWTcVOC<1, where cWT,cVOC are the viral trajectories clearance rate of the wild-type and VOC variants. We scaled the infectiousness profile for the VOCs shortening its time course by κ:
(6)hVOC(τi,τg) =hWT(κτi,κτg)

Where  hWT, hVOC <milestone-start /> <milestone-end /> are the joint bivariate distribution of incubation period and generation interval of the wild-type and VOC variants. Since the distribution of infectiousness is lognormal the scaling affects only one of the parameters of the distribution (the median). See Supplementary methods for full derivation. The resulting unmitigated generation-interval distribution then estimates the unmitigated infectiousness profile of new variants under a counterfactual scenario, in which behavioral and intervention effects remain the same as in the initial pandemic phase.”

(4) In order to make the manuscript more accessible to a general audience, please provide a clearer explanation of why short forward intervals are overrepresented in a growing epidemic. Consider including a diagram or simulation, as suggested by Reviewer 3.

We added an explanation to the paragraph in order to make it clearer:

“A cohort of individuals that develop symptoms on a given day is a sample of all individuals who have been previously infected. When the incidence of infection is increasing, recently infected individuals represent a bigger fraction of this population and thus are over-represented in this cohort. Therefore, we are more likely to encounter infected individuals with a short incubation period in this cohort compared to an unbiased sample. The forward serial-interval is calculated for a cohort of infectors who developed symptoms at the same time and therefore is sensitive to this bias. These dynamical biases are demonstrated using epidemic simulations by Park et al."

(5) Please address the impact of uncertainty in viral load trajectories on individual generation times, on the residual fraction, and on our ability to infer generation intervals for variants of concern using viral load trajectories. On a related note, please consider modifications to Figure 6a so that it is easier to visualize whether the viral load trajectory aligns well with the claim that 18% of transmission occurs >14d after infection.

Viral load trajectories data have potential for informing estimates of the infectiousness profile. However the relationship between viral load, culture positivity, symptom onset, and infectivity is complex and not well characterized. Due to this limitation we tried to use viral loads in a more limited way, extrapolating our results to variants of concerns (which lack unmitigated transmission data). Following the comment, we added a detailed discussion of the limitations of using viral loads as a proxy for infectiousness, including the variation of viral loads across individuals. We also added supplementary figures (Figure 6—figure supplements 1-2) to show the possible effect of an individual's viral loads in relation to the infectiousness and for comparison with new viral load and culture results (Chu et al. 2022; Killingley et al. 2022). As the viral load trajectories data for the different VOC is given only as a function of time from the onset of symptoms, it is not possible to directly link it to the fraction of transmission post 14 days from infection. We made changes to Figure 6 to clarify the possible connection of viral load with the TOST (time from symptoms onset to transmission) distribution and the resulting extrapolation to the unmitigated generation-interval distributions.

“SARS-CoV-2 viral load trajectories serve an important role in understanding the dynamics of the disease and modeling its infectiousness (Quilty et al. 2021; Cleary et al. 2021). Indeed, the general shapes of the mean viral load trajectories and culture positivity, based on longitudinal studies, are comparable with our estimated unmitigated infectiousness profile (Figure 6—figure supplements 1-2, comparison with (Chu et al. 2022; Killingley et al. 2022; Kissler et al. 2021)). However, the nature of the relationship between viral load, culture positivity, symptom onset, and real-world infectivity is complex and not well characterized. Therefore, the ability to infer infectiousness from viral load data is very limited, especially near the tail of infectiousness, several days following symptom onset and peak viral loads. Viral load models are usually made to fit the measurements during an initial exponential clearance phase and in many cases miss a later slow decay (Kissler et al. 2021). Furthermore, there is considerable individual-level variation in viral trajectories that isn’t accounted for in population-mean models (Kissler et al. 2021; Singanayagam et al. 2021). Other factors limiting the ability to compare generation-interval estimates with viral loads models are the variability of the incubation periods and its relation to the timing of the peak of the viral loads, and the great uncertainty and apparent non-linearity of the relation between viral loads and culture positivity (Jaafar et al. 2021; Jones et al. 2021). Due to these caveats and in order to avoid over interpretation of viral load data, we restrict our extrapolation of new VOCs’ infectiousness to a single parameter characterizing the viral duration of clearance.”

Reviewer #1 (Recommendations for the authors):1. The methods section is complete, but it might be easier to follow with more attention to organization, transitions, and maybe with additional subheadings. In particular, it would be helpful if key details like which parameters are being estimated, and which data you're fitting to, were easier to locate in this section.

We made some edits to the Methods section in order to make it more readable and clearer. We added subheadings for the various sections. Moreover, we added a section explaining the derivation of the basic reproduction number and clarified the section regarding the VOCs extrapolations.

2. The Introduction and Methods cover a lot of ground summarizing all the forms of bias and adjustment that go into producing an accurate unmitigated estimate, and it is currently a bit hard to keep track of all these details. It could be helpful to provide some sort of list, table, or summary paragraph to help readers keep track of all the forms of bias and adjustment that this analysis deals with, including references where appropriate. It would also be helpful to more clearly state that the main contribution of this study is to collect and apply all these statistical corrections to a carefully curated dataset.

Following the comment we added a table summarizing the main possible biases in inference of infectiousness profile from serial intervals data.

3. I got tripped up by this statement on page 12:"We find that our framework is able to properly reproduce the realized serial interval distribution given the growth rate in the early stages of the outbreak in Wuhan, China (Figure 3b)."Aren't the models fit to the SI data-meaning that we expect this result and should be alarmed by anything else? I think that this is just a wording issue and that what you're trying to say here is something like, "With or without the growth-rate adjustment, the model was able to fit the observed serial interval data well (Figure 3b)". But with the current phrasing, it sounds (at least to me) like this is being presented as some sort of independent validation of the model. For the same reason, I'd consider changing "estimated SI" to "fitted SI' in the Figure 3b legend.

We thank the reviewer for pointing out the confusing phrasing. We changed the phrasing as the reviewer suggest, and it now reads:

“The joint bivariate distribution and its marginal distributions are shown in Figure 3A. With or without the growth-rate adjustment, the model was able to fit the observed serial interval data well (Figure 3B)”.

Furthermore, the legend of Figure 3B was also changed accordingly to “Fitted SI”.

Reviewer #2 (Recommendations for the authors):As well as the broad comments made in the public review, I had the following comments:– "This dataset includes a total of 77 transmission pairs with a mean serial interval of 9.1 days (7.9-10.2 95% CIs) and a standard deviation of 5.2 days. This is substantially longer than the mean of 7.8 days suggested by Ali et al" – is it possible to quantify this difference statistically (e.g. with a test for difference in means between the samples)? Given a mean of 9.1 days and SD=5.2, it wouldn't seem implausible for a random subsample from this dataset to have a mean of 7.8?

As the reviewers pointed out, the 95% confidence interval of the mean serial interval estimate is 7.9-10.2 days, which overlaps with the one reported by Ali et al. (mean of 7.8 days, CI of 7-8.6 days). We now acknowledge this uncertainty in the main text:

“This dataset includes a total of 77 transmission pairs with a mean serial interval of 9.1 days (95% CI: 7.9-10.2), and a standard deviation of 5.2 days. Although this is substantially longer than the mean of 7.8 days (95% CI: 7-8.6 days) suggested by Ali et al. (Ali et al. 2020) for the early period of the epidemic, there is considerable uncertainty in both estimates with overlapping confidence intervals. Nonetheless, a lower mean serial interval estimated by Ali et al. likely reflects their decision to include infectors who developed symptoms up to January 22nd, who were already subject to effects on mitigation strategy”.

We do not think that quantifying the statistical significance will provide any more information, especially given that the confidence intervals overlap by a considerable amount. Nonetheless, we still find that the mean serial interval decreases consistently when we use later cut-off dates. We also note that our dataset incorporates several other transmission pairs taken from other sources.

– Could the authors clarify which formula they used from Wallinga and Lipsitch (2007) to calculate R0 from generation time, as the exact calculation will depend on assumptions about the distribution of generations etc? I presume the authors used an appropriate formulation but would be useful to state explicitly. The finding that the early R0 is similar despite a longer generation time seems a bit counter-intuitive, so it would be helpful to have some more discussion about what's happening here.

We added to the methods the formula for derivation of R_0_ using the distribution of the generation interval. See response for Essential Revisions 2.

The estimate of R_0_ is similar to the previous estimate mainly due to the use of the corrected growth rate of 0.1/d, as previous studies assumed shorter GI (which decreases R_0_) but higher growth rate (which increases R_0_).

“The basic reproduction number R_0_ estimates derived here are close to reported values from early in the epidemic value (Wu, Leung, and Leung 2020; Li et al. 2020; Chinazzi et al. 2020; Imai et al. 2020), despite the longer estimate for the generation-interval distribution. This is mainly due to using the corrected growth rate, which is considerably lower than previously assumed values (Tsang et al. 2020).”

– It would be useful to give some intuition about why changing the baseline incubation period had a limited effect on the results. Is this because the epidemic phase adjustment dominates in the calculation?

From the sensitivity analysis presented in Appendix 1—figure 10, changing of the incubation period distribution mainly affects the estimate of the correlation parameter (shorter incubation period causes a decrease in the correlation parameter). The adjustment for epidemic phase also doesn’t have a large effect on the results. Therefore, the cut-off date seems to be the dominant factor in our analysis, presumably meaning that mitigations have the largest effect on the generation interval distribution. We add the next paragraph to the discussion:

“Following the sensitivity analyzes to the cutoff date, the growth rate and the model of infectiousness, we can see which of the three biases described in Table 1 has the greatest effect. We conclude that the cutoff date seems to be the dominant factor in our analysis, presumably meaning that taking the effects of interventions into account is the most important for an accurate estimate of the generation interval distribution. Additional sensitivity analyses, such as to the assumed incubation period, also support this conclusion, as they show only a minor effect.”

– The methods for scaling the generation interval for other VOCs are described briefly in the caption to Figure 6, but it would be helpful to have the calculation given explicitly in the methods, so there is no ambiguity in terms like "ratio of the clearance's durations". Also in this figure, it's unclear where Α line is in B and C, so worth mentioning in the caption. Finally, I didn't follow this sentence: "The inset shows a zoom-in on the period of 12-24 days after exposure, a period in which there is a substantial difference between the current estimate and those from previous studies." Are the presented estimates not all new ones derived from the current study and viral shedding data.

We have now expanded our explanation for the extrapolation of the unmitigated generation intervals of the VOC in the Methods.

The extrapolation of the α and δ variants are extremely close, hence the hidden α line in the panels. We added in the figure caption:

“The extrapolated distributions for the α and δ variants are extremely close, hence the green line is hidden by the red line in panels b-d”.

The anomalous sentence in the figure caption was an editing error and has been removed. Thank you for pointing it out.

– I appreciate that not all of these studies were available at the time of submission, but it could be helpful to update the discussion to also place the results in the context of more recent viral culture duration from serial swabbing data (Chu et al., JAMA Int Med 2022) and/or shedding profiles in human challenge data (Killingley et al., Nature Med 2022).

Following the comment we added a supplemental figure (Figure 6—figure supplement 2) that shows comparison between the estimated profile of infection to these two recent studies. We also added a detailed discussion paragraph regarding the comparability of the results with viral load data and the major limitation of this kind of comparisons, see response to Essential Revisions 5.